# Promoting Drp1-mediated mitochondrial fission in midlife prolongs healthy lifespan of *Drosophila melanogaster*

Anil Rana[1], Matheus P. Oliveira[1,2], Andy V. Khamoui[3,6], Ricardo Aparicio[1], Michael Rera [1,7], Harry B. Rossiter [3,4] & David W. Walker[1,5]

The accumulation of dysfunctional mitochondria has been implicated in aging, but a deeper understanding of mitochondrial dynamics and mitophagy during aging is missing. Here, we show that upregulating Drp1—a Dynamin-related protein that promotes mitochondrial fission—in midlife, prolongs *Drosophila* lifespan and healthspan. We find that short-term induction of Drp1, in midlife, is sufficient to improve organismal health and prolong lifespan, and observe a midlife shift toward a more elongated mitochondrial morphology, which is linked to the accumulation of dysfunctional mitochondria in aged flight muscle. Promoting Drp1-mediated mitochondrial fission, in midlife, facilitates mitophagy and improves both mitochondrial respiratory function and proteostasis in aged flies. Finally, we show that autophagy is required for the anti-aging effects of midlife Drp1-mediated mitochondrial fission. Our findings indicate that interventions that promote mitochondrial fission could delay the onset of pathology and mortality in mammals when applied in midlife.

[1] Department of Integrative Biology and Physiology, University of California, Los Angeles, CA 90095, USA. [2] Instituto de Bioquimica Medica Leopoldo de Meis, Universidade Federal do Rio de Janeiro, Cidade Universitaria, Rio de Janeiro 21941-590, Brazil. [3] Division of Respiratory & Critical Care Physiology & Medicine, Los Angeles Biomedical Research Institute at Harbor-UCLA Medical Center, Torrance, CA 90502, USA. [4] Faculty of Biological Sciences, University of Leeds, Leeds LS2 9JT, UK. [5] Molecular Biology Institute, University of California, Los Angeles, CA 90095, USA. [6] Present address: Department of Exercise Science and Health Promotion, Florida Atlantic University, Boca Raton, FL 33431, USA. [7] Present address: Laboratory of Degenerative Processes, Stress and Aging, Université Paris Diderot, Paris 75013, France. Correspondence and requests for materials should be addressed to D.W.W. (email: davidwalker@ucla.edu)

Mitochondrial dysfunction is a key hallmark of aging and has been linked to numerous age-onset pathologies[1–3]. Therefore, identifying interventions that could improve mitochondrial homeostasis when targeted to aged animals would be highly desirable toward the goal of prolonging healthspan. A growing body of data support the idea that autophagy has an important anti-aging role[4, 5]. However, the relevant autophagic cargo in the context of aging remains elusive. Mitochondrial autophagy (mitophagy) is a type of cargo-specific autophagy[6, 7], which mediates the removal of dysfunctional mitochondria. The molecular mechanisms of mitophagy have been elucidated in some detail in recent years[8]. Upon loss of inner mitochondrial membrane potential, PINK1, a mitochondrial kinase, is selectively stabilized on the surface of the dysfunctional mitochondrion, leading to the recruitment of the E3 ubiquitin ligase Parkin[9]. Upon mitochondrial recruitment, Parkin ubiquitinates mitochondrial outer membrane proteins[10–14] and induces the autophagic elimination of the dysfunctional mitochondrion[6–8].

Recent studies in mammals, including humans, have reported an age-related decline in mitophagy[15, 16]. Moreover, impairment of mitophagy recapitulates the age-related accumulation of mitochondria in Caenorhabditis elegans[17]. These findings suggest that the mitophagy pathway may represent a therapeutic target to counteract aging. Consistent with this idea, overexpression of Parkin delays the onset of molecular markers of aging and prolongs lifespan in Drosophila[18]. Furthermore, dietary urolithin A (UA) treatment induces mitophagy, prevents the age-related accumulation of dysfunctional mitochondria and prolongs C. elegans lifespan[19]. UA treatment also improves exercise capacity in rodent models of age-related decline of muscle function[19]. Together, these findings support the idea that impaired mitophagy is a significant underlying factor in the accumulation of dysfunctional mitochondria in aged animals contributing to organismal health decline and mortality. However, a major unanswered question remains: why does mitophagy decline in aged animals?

Mitochondrial dynamics (fission and fusion) and mitophagy are closely related[7, 8, 20, 21]. Mitochondrial fission and fusion processes are both mediated by large guanosine triphosphatases (GTPases) in the dynamin family[22, 23]. Mitofusin (Mfn) proteins mediate fusion of the mitochondrial outer membrane, while mitochondrial fission, conversely, requires Dynamin-related protein 1 (Drp1)[22, 24]. Several studies indicate that an important event preceding mitophagy is the Parkin-mediated turnover of Mfn leading to a shift in the balance of mitochondrial dynamics toward decreased fusion/increased fission[7, 20]. In yeast, the mitochondrial fission protein, Dnm1, homologous to Drp1, is required for certain forms of mitophagy[25–27]. Together, these findings support the model that mitochondrial fission can promote the segregation of damaged mitochondria and facilitate their clearance by mitophagy[7, 11, 12, 28, 29]. Critically, however, the interplay between mitochondrial dynamics and mitophagy during aging remains poorly understood.

The anti-aging effects of Parkin overexpression in Drosophila[18] and UA treatment in C. elegans[19] are both associated with an increase in mitochondrial fission. Critically, however, the question of whether an increase in mitochondrial fission alone is sufficient to prolong lifespan and/or improve mitochondrial function in an aged animal has not been addressed. Here, we show that inducing Drp1-mediated mitochondrial fission, in midlife, increases lifespan and improves multiple markers of health in aged Drosophila. Remarkably, we show that a transient induction of Drp1, for 7 days, in midlife is sufficient to prolong lifespan. Studying aging flight muscle, we find that a midlife shift toward a more elongated, less circular mitochondrial morphology is linked to the accumulation of dysfunctional mitochondria.

Short-term, midlife Drp1 induction restores mitochondrial morphology to a youthful state, improves mitochondrial respiratory function and reduces mitochondrial reactive oxygen species (ROS) levels. Importantly, midlife Drp1 induction facilitates mitophagy and improves proteostasis in aged flies. Finally, we show that disruption of Atg1, a core autophagy gene, inhibits the anti-aging, prolongevity effects of midlife Drp1 induction. Our findings indicate that transient, midlife interventions that promote mitochondrial fission could delay the onset of frailty and mortality in aging mammals.

## Results

**Midlife Drp1 induction prolongs lifespan and healthspan.** In previous work, we showed that long-lived Parkin overexpressing flies display an increase in mitochondrial fission[18]. To test for a causal role for mitochondrial fission in lifespan extension, we sought to independently promote mitochondrial fission by upregulating Drp1 in aging flies. To do so, we used the ubiquitous daughterless-Gene-Switch (daGS) driver line to activate a UAS-Drp1 transgene created by[30]. Western blot and immuno-fluorescence analysis confirmed a dose-dependent and RU486-dependent induction of the Drp1 transgene (Supplementary Fig. 1a–d). Next, we confirmed that Drp1 induction confers a shift in mitochondrial dynamics toward increased fission in adult flies in both muscle and brain tissue (Fig. 1a–c). Next, we set out to examine the impact of temporally-defined shifts in mitochondrial dynamics on fly lifespan. Upregulation of Drp1 in early adulthood (days 0–30) or throughout adulthood had no significant impact on longevity (Fig. 1d, Supplementary Table 1). However, upregulating Drp1 from midlife (day 30) onwards significantly extended lifespan in both male and female flies (Fig. 1e, Supplementary Fig. 1e, f; Supplementary Table 1). Using an independently generated UAS-Drp1 transgene[31], we confirmed that midlife Drp1 induction prolongs lifespan (Supplementary Fig. 1g; Supplementary Table 1). To further examine the impact of manipulating mitochondrial dynamics in midlife, we examined the impact of RNAi of Drosophila (d)Mfn on lifespan. Importantly, we observed that RNAi of dMfn from midlife onwards also extends both mean and maximum lifespan (Fig. 1f; Supplementary Table 1). Taken together, our findings demonstrate that promoting mitochondrial fission/inhibiting fusion from midlife onwards prolongs fly lifespan. To further validate our findings, we examined the impact of inhibiting mitochondrial fission/promoting mitochondrial fusion in midlife on fly lifespan. Importantly, we observed that expression of a dominant-negative Drp1 (Drp1[K38A]) transgene[30] from midlife onwards shortens lifespan (Supplementary Fig. 1h; Supplementary Table 1) and upregulation of dMfn in midlife also shortens lifespan (Supplementary Fig. 1i; Supplementary Table 1). To refine the tissue-specific requirements involved in Drp1-mediated lifespan extension, we used the pan-neuronal Elav–Gene-Switch (ElavGS) driver line to increase Drp1 expression specifically in neurons[32] and 5966-GS[33] to increase Drp1 gene expression in the intestine. Upregulating Drp1 specifically in neurons or the intestine from midlife onwards is sufficient to prolong fly lifespan (Supplementary Fig. 1j, k; Supplementary Table 1).

To refine the temporal requirements involved, we tested whether a transient induction of Drp1 was sufficient to prolong lifespan. We found that short-term induction of Drp1, from day 30 to 37, produced a robust increase in both mean and maximum lifespan (Fig. 1g, Supplementary Fig. 1l; Supplementary Table 1). Importantly, we find that Drp1 mRNA levels do not remain elevated following short-term midlife induction (Fig. 1h). RU486 had no impact on mitochondrial morphology, Drp1 mRNA levels or longevity in control flies when fed at any time period

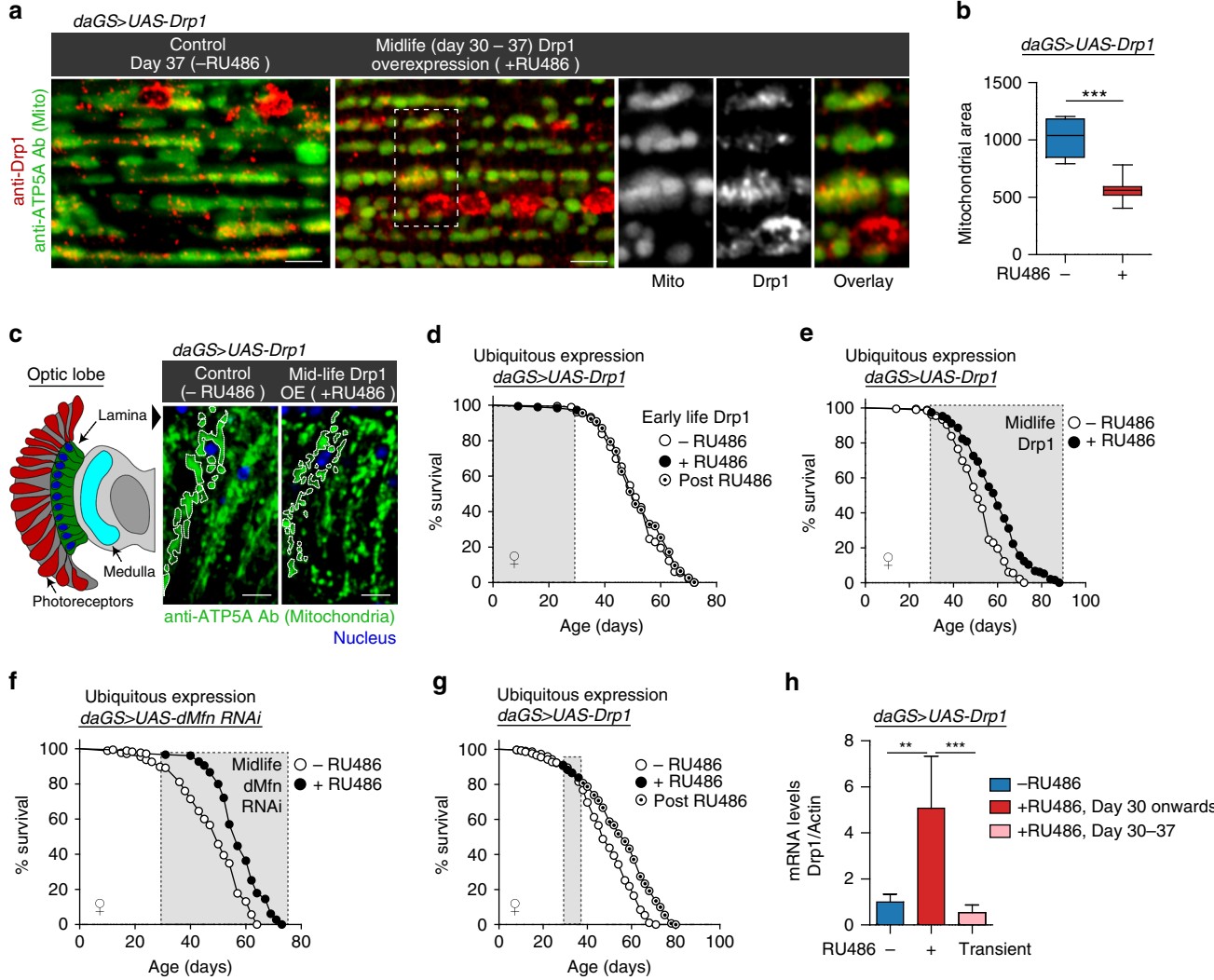

**Fig. 1** Midlife Drp1 induction extends lifespan. **a**, **b** Immunostaining of indirect flight muscles **a** from 37 day old *daGS > UAS-Drp1* females with or without RU486-mediated transgene induction for 7 days from day 30 to day 37, showing mitochondria (*green channel*, anti-ATP5a) and Drp1 (*red channel*, anti-Drp1). *Scale bar* is 5 μm. Quantification of mitochondrial size **b**; n = 7 individual thoraces; ***p < 0.001; two-tailed unpaired *t*-test. **c** Immunostaining of optic lobe/brain from 37 day old *daGS > UAS-Drp1* females with or without RU486-mediated transgene induction for 7 days from day 30 to day 37, showing mitochondria (*green channel*, anti-ATP5a) and nuclei (*blue channel*, TO-PRO-3). *Scale bar* is 5 μm. **d** Survival curves of *daGS > UAS-Drp1* females with or without RU486-mediated transgene induction from day 1 to day 30. The *shaded area* indicates the duration of *Drp1* induction. p = 0.42, log-rank test; n > 179 flies. **e** Survival curves of *daGS > UAS-Drp1* females with or without RU486-mediated transgene induction from day 30 onwards. The *shaded area* indicates the duration of *Drp1* induction. p < 0.0001, log-rank test; n > 179 flies. **f** Survival curves of *daGS > UAS-dMfn-RNAi* females with or without RU486-mediated transgene induction from day 30 onwards. The *shaded area* indicates the duration of *dMfn* RNAi. p < 0.0001, log-rank test; n > 175 flies. **g** Survival curves of *daGS > UAS-Drp1* females with or without RU486-mediated transgene induction from day 30 to day 37. The *shaded area* indicates the duration of *Drp1* induction. p < 0.0001, log-rank test; n > 291 flies. **h** qPCR analyses of *Drp1* mRNA levels on day 44 in *daGS > UAS-Drp1* females with or without RU486-mediated transgene induction from day 30 to day 37. n = 5 biological replicates with three individual flies per replicate; ***p < 0.001 and **p < 0.01; one-way ANOVA/Bonferroni's multiple comparisons test. Boxplots **b** display the first and third quartile, with the horizontal *bar* at the median and *whiskers* showing the most extreme data point, which is no more than 1.5 times the interquartile range from the box. *Bars* **h** depict mean ± s.d. RU486 was provided in the media at a concentration of 25 μg/ml for **a–e**, **g**, **h** and 50 μg/ml for **f**

(Supplementary Figs 1m–p and 3d, e). Because ubiquitous, midlife Drp1 induction in female flies resulted in the most pronounced extension of lifespan, this paradigm was used in all further experiments.

To better understand the impact of Drp1-mediated mitochondrial fission in midlife on organismal health, we examined a number of markers of healthspan and behavior in aged flies following 7 days of Drp1 induction. As a reduction in food intake can modulate lifespan, we tested whether midlife Drp1 induction affects feeding behavior. Using a capillary feeding assay[34], we failed to observe alterations in feeding behavior upon short-term,

midlife Drp1 induction (Fig. 2a). It is important to determine whether interventions that prolong lifespan also extend healthspan or simply prolong a period of frailty. Short-term, midlife Drp1 induction conferred an increase in spontaneous physical activity (Fig. 2b and c). Importantly, this increase in physical activity was confined to day-time activity, as opposed to night time restlessness. Moreover, short-term, midlife Drp1 induction produced a significant improvement in an endurance exercise paradigm (Fig. 2d). The ability to withstand extrinsic stress is a marker of organismal health that declines with age. Short-term, midlife Drp1 induction improved survival when maintained on

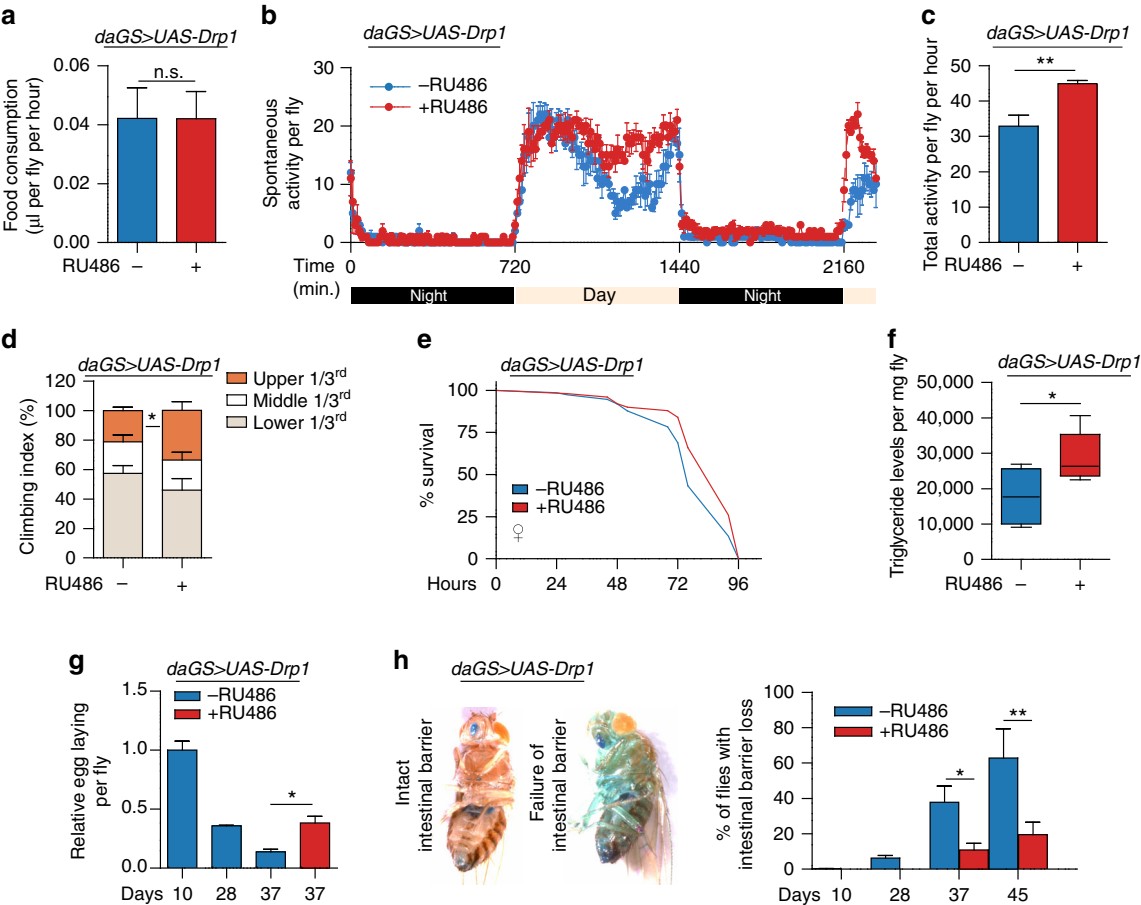

**Fig. 2** Midlife Drp1 induction delays age-onset pathology and prolongs healthspan. **a** Capillary feeding assay (CAFE) of 37 day old *daGS > UAS-Drp1* females with or without RU486-mediated transgene induction from day 30 to day 37. $n = 8$ vials of 10 flies per condition; $p > 0.05$ and is non-significant (n.s.); two-tailed unpaired *t*-test. **b**, **c** Spontaneous physical activity **b** of 37 day old *daGS > UAS-Drp1* females with or without RU486-mediated transgene induction from day 30 to day 37. Quantification of total activity per fly per hour **c** from spontaneous activity graphs. $n = 3$ vials of 10 flies per condition; **$p < 0.01$; two-tailed unpaired *t*-test. **d** Climbing index as a measure of endurance of 37 day old *daGS > UAS-Drp1* females with or without RU486-mediated transgene induction from day 30 to day 37. $n = 90$ flies per condition; *$p < 0.05$; two-tailed Mann–Whitney test. **e** Survival curves without food of 37 day old *daGS > UAS-Drp1* females with or without RU486-mediated transgene induction from day 30 to day 37. $p < 0.01$; log-rank test; $n = 100$ flies. **f** Whole body lipid stores of 37 day old *daGS > UAS-Drp1* females with or without RU486-mediated transgene induction from day 30 to day 37. $n = 5$ biological replicates with three flies per replicate; *$p < 0.05$; two-tailed unpaired *t*-test. **g** Fecundity of *daGS > UAS-Drp1* females with or without RU486-mediated transgene induction from day 30 to day 37. $n = 630$ flies on day 10; *$p < 0.05$; one-way ANOVA/Bonferroni's multiple comparisons test. **h** Intestinal integrity during aging of *daGS > UAS-Drp1* females with or without RU486-mediated transgene induction since midlife (day 30) onwards. $n = 448$ flies on day 10; **$p < 0.01$, *$p < 0.05$; one-way ANOVA/Bonferroni's multiple comparisons test. *Bars* **a**, **c** and **d** depict mean ± s.d. and *bars* **g**, **h** depict mean ± s.e.m. Boxplots **f** display the first and third quartile, with the horizontal *bar* at the median and *whiskers* showing the most extreme data point, which is no more than 1.5 times the interquartile range from the box. RU486 was provided in the media at a concentration of 25 μg/ml

an agar-only diet to induce starvation in aged flies (Fig. 2e). This improvement in starvation resistance was linked to increased triglyceride (TAG) stores (Fig. 2f). Several interventions that promote longevity, such as dietary restriction, are frequently associated with reproductive tradeoffs[35]. In contrast, short-term, midlife Drp1 induction increased fecundity in aged flies (Fig. 2g).

Aging is associated with a loss of tissue homeostasis, resulting in a decline in organ function. Recent work has shown that loss of intestinal barrier function is an evolutionarily conserved biomarker of aging[36–40]. To determine whether promoting mitochondrial fission in midlife can delay the onset of intestinal pathology, we examined intestinal barrier function in aging flies with and without midlife Drp1 induction. Loss of intestinal integrity can be assayed in living flies by monitoring the presence of non-absorbed dyes outside of the digestive tract post-feeding[36, 41] (Fig. 2h). Remarkably, we observed a delay in the onset of intestinal barrier dysfunction upon Drp1 induction

from day 30 onwards (Fig. 2h), indicating a delay in intestinal aging at the tissue level. Collectively, these data demonstrate that promoting Drp1-mediated mitochondrial fission in midlife improves healthspan and delays the onset of pathology linked to aging. Consistent with these findings, we observed that inhibiting mitochondrial fission or promoting mitochondrial fusion in midlife accelerates age-onset pathology. More specifically, either Drp1[K38A] expression (Supplementary Fig. 2g) or upregulation of *dMfn* (Supplementary Fig. 2h) in midlife confers early-onset intestinal barrier dysfunction. Feeding RU486 to control flies did not alter feeding behavior, spontaneous physical activity, loss of intestinal integrity, fecundity, or starvation sensitivity (Supplementary Fig. 2a–f).

**Midlife Drp1 induction improves mitochondrial function in aged flies**. To better understand the interplay between mitochondrial dynamics and aging, we used immunofluorescence (IF)

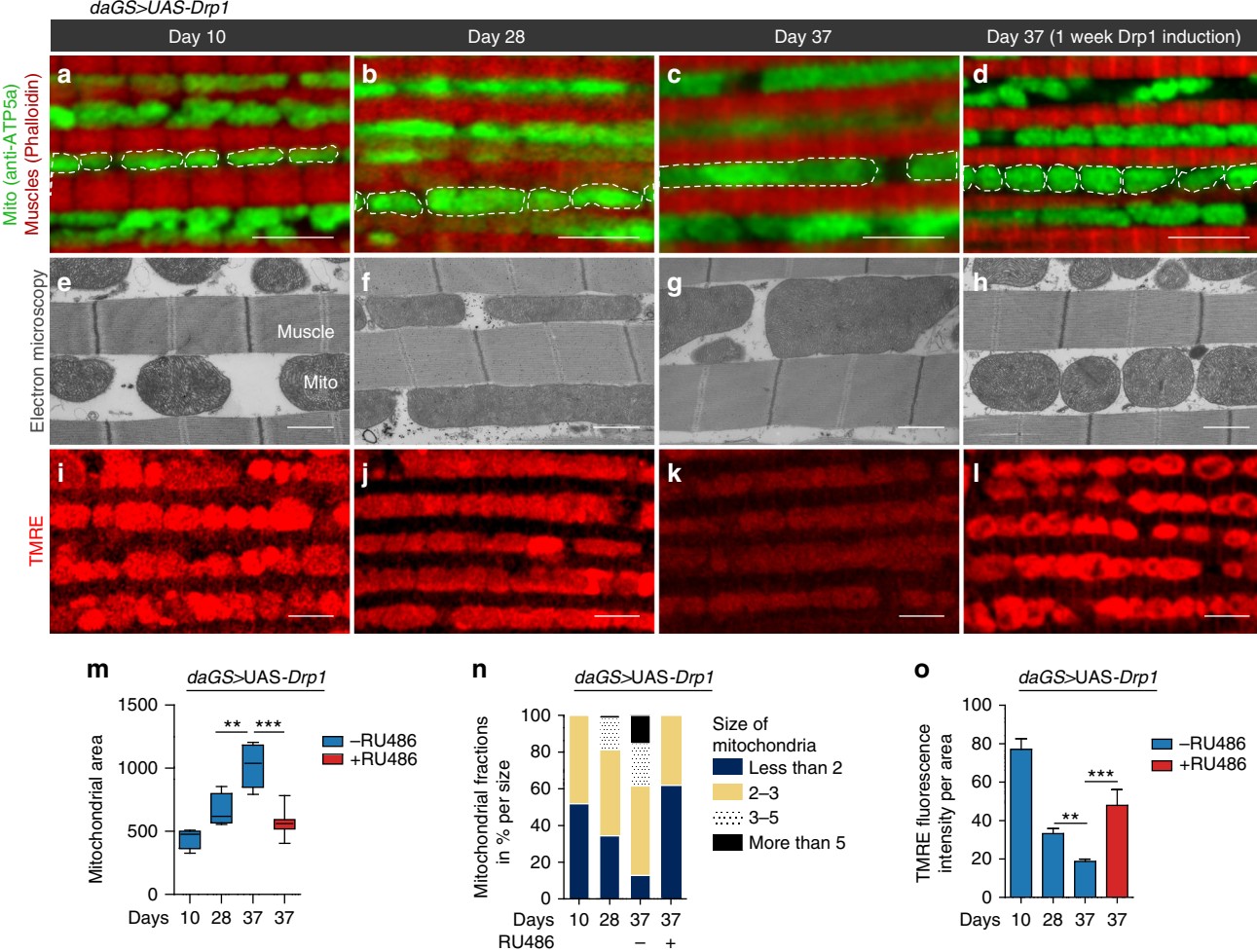

**Fig. 3** Midlife Drp1 induction rejuvenates mitochondrial morphology and function. **a–d** Immunostaining of indirect flight muscles from day 10, 28 and 37 *daGS > UAS-Drp1* females with or without RU486-mediated transgene induction from day 30 to day 37 showing mitochondria (*green channel*, anti-ATP5a) and muscles (*red channel*, rhodamine staining for F-actin). *Scale bar* is 5 μm. **e–h** Electron micrograph of indirect flight muscles from day 10, 28 and 37 *daGS > UAS-Drp1* females with or without RU486-mediated transgene induction from day 30 to day 37, showing mitochondria and muscles. *Scale bar* is 1 μm. **i–l** Staining of indirect flight muscles from day 10, 28, and 37 *daGS > UAS-Drp1* females with or without RU486-mediated transgene induction from day 30 to day 37 showing TMRE fluorescence. *Scale bar* is 5 μm. **m** Quantification of mitochondrial size in indirect flight muscles from day 10, 28, and 37 *daGS > UAS-Drp1* females with or without RU486-mediated transgene induction from day 30 to day 37 as shown in **a–d**. $n = 4$–7 flies; ***$p < 0.001$, **$p < 0.01$; one-way ANOVA/Bonferroni's multiple comparisons test. **n** Quantification of relative mitochondrial size in indirect flight muscles from day 10, 28, and 37 *daGS > UAS-Drp1* females with or without RU486-mediated transgene induction from day 30 to day 37 as shown in **e–h**. Data are represented as mitochondrial fractions in percentages per size. $n = 3$ flies per condition. **o** Quantification of mitochondrial membrane potential measured by TMRE staining as shown in **i–l** from day 10, 28, and 37 *daGS > UAS-Drp1* females with or without RU486-mediated transgene induction from day 30 to day 37. $n = 10$–19 flies; ***$p < 0.001$, **$p < 0.01$; Kruskal–Wallis test/Dunn's multiple comparisons test. Boxplots **m** display the first and third quartile, with the horizontal *bar* at the median and *whiskers* showing the most extreme data point, which is no more than 1.5 times the interquartile range from the box. Bars **o** depict mean ± s.e.m. RU486 was provided in the media at a concentration of 25 μg/ml

microscopy and electron microscopy (EM) to examine age-related alterations in mitochondrial morphology in flight muscle. Using both approaches, we observed that in midlife (day 28) (Fig. 3b, f) the mitochondria are more elongated and less circular than in young muscle tissue (10 day old) (Fig. 3a, e). This shift in mitochondrial morphology becomes more pronounced at day 37 (Fig. 3c, g) suggesting an increase in fusion/decrease in fission. To examine the generality of this finding, we examined age-related alterations in mitochondrial morphology in an independent control laboratory strain, $w^{dahomey}$, and confirmed a midlife shift toward a more elongated mitochondrial morphology in aged flight muscle (Supplementary Fig. 3a, b). To determine whether this age-related shift in mitochondrial dynamics was linked to altered expression of Drp1, we examined Drp1 expression levels during aging. We observed that an age-related decline in Drp1

expression accompanies the shift toward a more elongated, less fragmented mitochondrial morphology (Supplementary Fig. 3c).

To gain insight into whether this shift in mitochondrial morphology is linked to altered mitochondrial function, we examined mitochondrial membrane potential using the potentiometric dye TMRE. Importantly, we observe that the accumulation of elongated mitochondria is linked to a significant decrease in TMRE fluorescence in aged flight muscle (Fig. 3i–k). Remarkably, we find that short-term, midlife induction (day 30–37) of Drp1 restores mitochondrial morphology, including cristae ultrastructure, to a more youthful state (compare Fig. 3d with 3a–c and Fig. 3h with 3e–g); quantification in Fig. 3m, n). Furthermore, we find that midlife induction of Drp1 restores a fragmented mitochondrial network, containing highly active mitochondria reminiscent of young tissue (compare Fig. 3l with 3i–k;

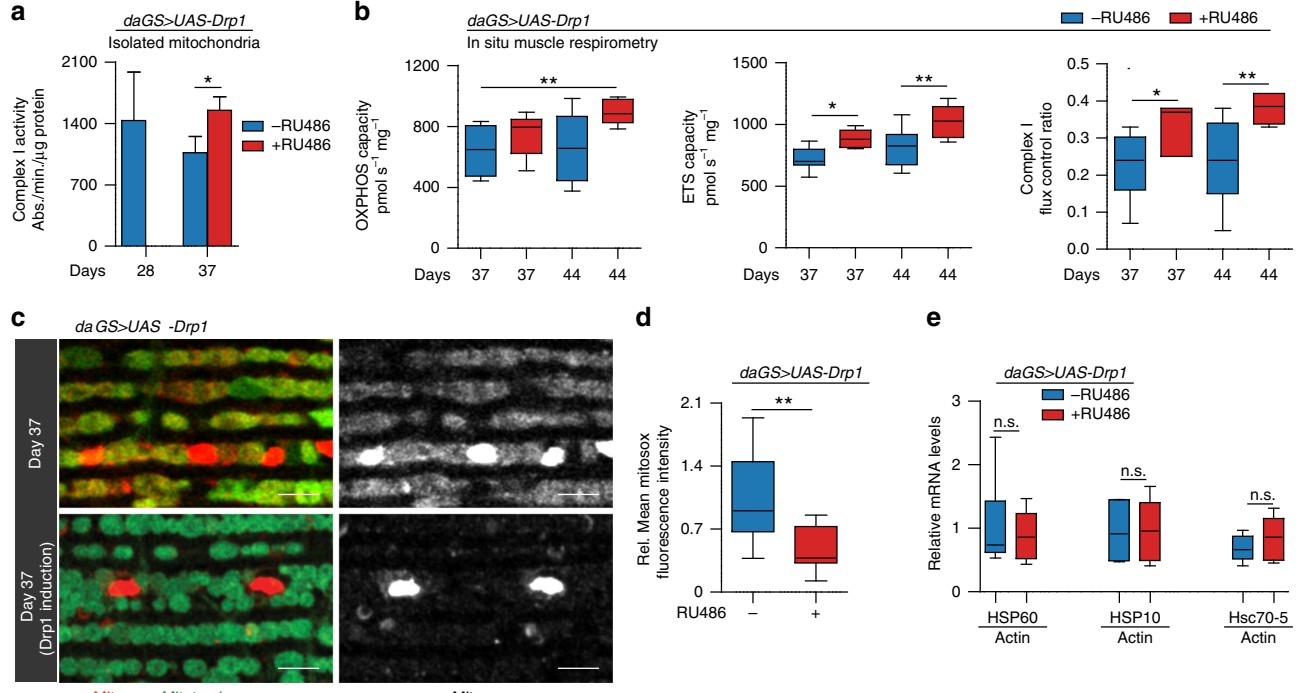

**Fig. 4** Midlife Drp1 induction improves mitochondrial respiratory function. **a**, **b** Quantification of markers of mitochondrial activity in 28, 37, and 44 day old *daGS > UAS-Drp1* females with or without RU486-mediated transgene induction since midlife (day 30 onwards). Complex I activity measurements **a** in isolated mitochondrial pellet from 28 to 37 day old adult females. $n = 5$ biological replicates with five flies per replicate; $**p < 0.01$; two-tailed unpaired *t*-test. **b** In situ respirometry of permeabilized muscle bundles from 37 to 44 day old adult females to assess the capacity for oxidative phosphorylation (OXPHOS) and Electron Transport System (ETS) flux, and the flux control ratio of Complex I by rotenone inhibition. $n = 6-8$ biological replicates with 2 flies per replicate; $**p < 0.01$ and $*p < 0.05$; one-way ANOVA/Bonferroni's multiple comparisons test. **c**, **d** Staining of indirect flight muscles **c** from 37 day old *daGS > UAS-Drp1* females with or without RU486-mediated transgene induction from day 30 to day 37, showing mitochondria (*green channel*, Mitotracker green staining) and levels of superoxide (*red channel*, staining with MitoSOX reagent). *Scale bar* is 5 μm. Quantification of free superoxide radicals **d**; $n = 11-16$ biological replicates; $**p < 0.01$; two-tailed unpaired *t*-test. **e** qPCR analyses of *Hsp60*, *Hsp10*, and *mtHsp70* (*Hsc70-5*) on day 37 in *daGS > UAS-Drp1* females with or without RU486-mediated transgene induction from day 30 to day 37. $n = 5$ biological replicates with 3 flies per replicate; $p > 0.05$ and is non-significant (n.s.); two-tailed unpaired *t*-test. Bars **a** depict mean ± s.d. Boxplots **b**, **d** and **e** display the first and third quartile, with the horizontal *bar* at the median and *whiskers* showing the most extreme data point, which is no more than 1.5 times the interquartile range from the box. RU486 was provided in the media at a concentration of 25 μg/ml

quantification Fig. 3o). Together, these findings indicate that the midlife shift in mitochondrial morphology, toward more elongated, less circular mitochondria, contributes to the accumulation of depolarized mitochondria in aged flight muscle. Critically, upregulating Drp1 expression in midlife, for 7 days, rejuvenates mitochondrial morphology and function. RU486 had no impact on mitochondrial morphology or TMRE fluorescence in control flies (Supplementary Fig. 3d–g).

Next, we set out to examine the impact of midlife Drp1 induction on additional markers of mitochondrial health and function in aged flight muscle. First, we observed a significant increase in complex I enzymatic activity following short-term (7 days) induction of Drp1 (Fig. 4a). To build upon this finding, oxidative phosphorylation (OXPHOS) and electron transport system (ETS) capacities were measured, in permeabilized muscle bundles in situ, using high-resolution respirometry. We observed a decline in OXPHOS activity and ETS capacity in aged flies (Supplementary Fig. 4a). Upregulation of Drp1 throughout adulthood did not significantly improve mitochondrial respiratory activity (Supplementary Fig. 4b). However, promoting Drp1-mediated mitochondrial fission from midlife onwards resulted in an increase in OXPHOS capacity at day 44, and an increase in overall ETS capacity and the flux control ratio for Complex I on days 37 and 44 (Fig. 4b). In addition, we validated that promoting Drp1-mediated mitochondrial fission from midlife onwards

improved mitochondrial function in brain tissue. More specifically, we observed that 7 days of Drp1 induction in midlife (day 30–37) resulted in an increase in OXPHOS capacity, an increase in overall ETS capacity and the flux control ratio for Complex I in head samples (Supplementary Fig. 4c). Next, we set out to determine whether short-term, midlife Drp1 induction impacts mitochondrial reactive oxygen species (ROS) levels. Following 7 days of Drp1 induction in midlife, we observed reduced mitochondrial ROS levels in aged flight muscle (Fig. 4c, d, Supplementary Fig. 4d). Taken together, our data show that promoting mitochondrial fission in midlife improves multiple markers of mitochondrial function and reduces mitochondrial ROS levels. RU486 had no significant effect on Complex I enzymatic activity or mitochondrial ROS levels in control flies (Supplementary Fig. 4e–g).

In recent years, considerable attention has been paid to the concept of mitohormesis[42]. Indeed, studies in diverse organisms have shown that certain perturbations that impair mitochondrial function can promote longevity through mechanisms that may involve elevated mitochondrial ROS and induction of the mitochondrial unfolded protein response (UPR[mt])[43–47]. As noted above, however, short-term, midlife Drp1 induction leads to reduced mitochondrial ROS levels (Fig. 4c, d). In addition, we failed to detect an induction of UPR[mt] genes following midlife Drp1 induction (Fig. 4e). These data indicate that midlife

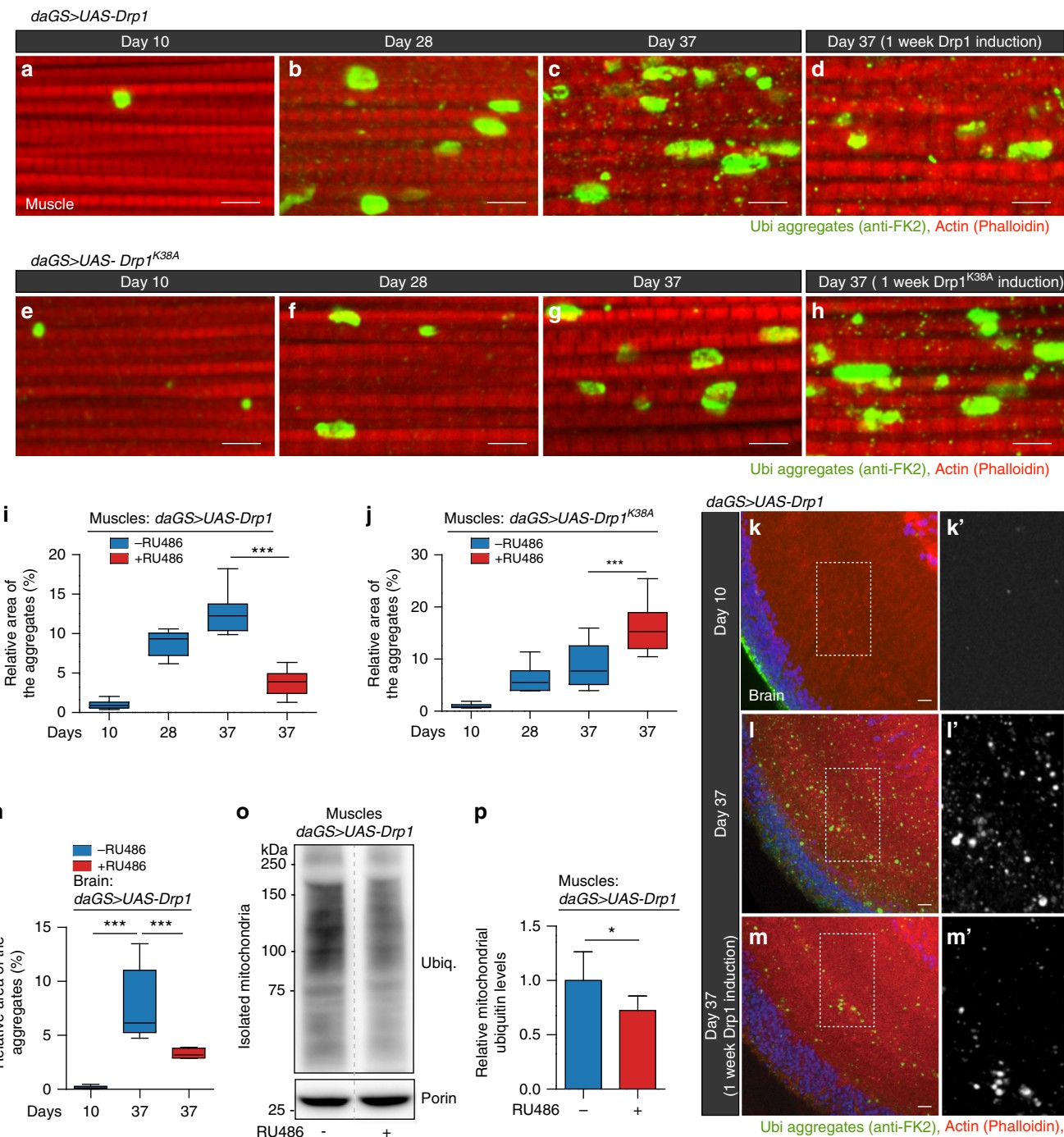

**Fig. 5** Midlife Drp1 induction improves proteostasis in aged flies. **a–d** Immunostaining of indirect flight muscles from day 10, 28, and 37 *daGS > UAS-Drp1* females with or without RU486-mediated transgene induction from day 30 to day 37, showing protein polyubiquitinated aggregates (*red channel*, muscles stained with phalloidin/F-actin and *green channel*, antipolyubiquitin). **e–h** Immunostaining of indirect flight muscles from day 10, 28, and 37 *daGS > UAS-Drp1$^{K38A}$* females with or without RU486-mediated transgene induction from day 30 to day 37, showing protein polyubiquitinated aggregates (*red channel*, muscles stained with phalloidin/F-actin and *green channel*, antipolyubiquitin). **i** Quantification of polyubiquitin aggregates in muscle as shown in **a–d**. $n = 9$–11 flies; ***$p < 0.001$; one-way ANOVA/Bonferroni's multiple comparisons test. **j** Quantification of polyubiquitin aggregates in muscle as shown in **e–h**. $n = 9$–14 flies; ***$p < 0.001$; one-way ANOVA/Bonferroni's multiple comparisons test. **k–m** Immunostaining of optic lobe/brain from day 10 and day 37 *daGS > UAS-Drp1* females with or without RU486-mediated transgene induction from day 30 to day 37, showing protein polyubiquitinated aggregates (*red channel* stained with phalloidin/F-actin; *green channel*, antipolyubiquitin and *blue channel*, nuclei stained with TO-PRO-3). **k'–m'** Insets from **k–m**, respectively. **n** Quantification of polyubiquitin aggregates in optic lobe/brain as shown in **k**, **m**. $n = 6$–13 flies; ***$p < 0.001$; one-way ANOVA/Bonferroni's multiple comparisons test. **o**, **p** Western blot **o** detection of total ubiquitin-conjugated proteins in isolated mitochondria from day 37 *daGS > UAS-Drp1* females with or without RU486-mediated transgene induction from day 30 to day 37. Densitometry of ubiquitin blots **p** from mitochondrial pellet. $n = 6$ replicates (25 flies per replicate); ***$p < 0.001$; two-tailed Mann–Whitney test. Boxplots **i**, **j** and **n** display the first and third quartile, with the horizontal *bar* at the median and *whiskers* showing the most extreme data point, which is no more than 1.5 times the interquartile range from the box. *Bars* **p** depict mean ± s.d. *Scale bar* is 5 μm. RU486 was provided in the media at a concentration of 25 μg/ml

Drp1-mediated improvements in mitochondrial function and organismal health are not mediated via mitohormesis. RU486 had no impact on UPR$^{mt}$ gene expression in control flies (Supplementary Fig. 4h).

**Midlife Drp1 induction improves proteostasis in aged flies**. Mitochondrial dysfunction and loss of protein homeostasis (proteostasis) are two key hallmarks of aging[1]. However, the relationship between these two hallmarks of aging is not well understood. As noted above, we find that promoting Drp1-mediated mitochondrial fission in midlife improves mitochondrial function in aged flies. To determine whether midlife Drp1 induction could impact proteostasis in aged animals, we characterized the deposition of protein aggregates in aged flight muscle by IF microscopy. As previously reported[18, 48], we observed that *Drosophila* flight muscles accumulate aggregates of ubiquitinated proteins during aging (Fig. 5a–c), consistent with a loss of proteostasis. Remarkably, short-term, midlife induction (days 30–37) of Drp1 resulted in reduced levels of protein aggregates in aged muscle (Fig. 5d, quantification in Fig. 5i) and aged brain tissue (Fig. 5k–m, quantification in Fig. 5n). To further validate our findings, we examined the impact of inhibiting mitochondrial fission/promoting mitochondrial fusion in midlife on proteostasis. Importantly, we observed that expressing a dominant-negative Drp1 (Drp1$^{K38A}$) transgene from midlife onwards impairs proteostasis (Fig. 5e–h, j) and upregulation of *dMfn* in midlife also impairs proteostasis (Supplementary Fig. 5a, b).

Recently, it has been reported that there is an over-representation of mitochondrial proteins in the insoluble protein fraction from aged animals[49, 50]. Interestingly, we observed that short-term, midlife Drp1 induction leads to reduced levels of ubiquitinated proteins in the mitochondrial fraction from aged flies (Fig. 5o, p, Supplementary Fig. 5e–g). Taken together, our findings reveal that the midlife shift toward more elongated mitochondria, in flight muscle, contributes to age-onset proteotoxicity. Critically, short-term, midlife induction of Drp1 improves proteostasis in aged animals. RU486 had no significant impact on the accumulation of protein aggregates in control flies (Supplementary Fig. 5c, d).

**Midlife Drp1 induction facilitates mitophagy in aged flies**. To further understand the impact of promoting mitochondrial fission in midlife on mitochondrial homeostasis, we examined markers of mitochondrial content following Drp1 induction. Short-term, midlife induction of Drp1 (days 30–37) resulted in lower levels of some respiratory chain subunit proteins (Fig. 6a, b and Supplementary Fig. 6a) and lower mitochondrial DNA levels in aged flight muscle (Fig. 6c, d). Furthermore, short-term midlife induction of Drp1 correlated with lower levels of dMfn, a degradation target of Parkin[8], in the mitochondrial fraction from aged flies (Fig. 6e, f and Supplementary Fig. 6d). A possible explanation for these findings is that midlife Drp1 induction facilitates mitophagy and/or promotes selective turnover of some respiratory chain subunits. Indeed, recent work has shown that some mitochondrial respiratory chain proteins appear to be selectively routed for autophagosomal degradation by the PINK1-Parkin pathway[51]. To test the idea that midlife Drp1 induction facilitates mitophagy, we examined aged flight muscle for co-localization of autophagy markers with mitochondria. Short-term, midlife Drp1 induction (day 30–33) resulted in a significant increase in the co-localization of Atg8a/LC3 with mitochondria (Fig. 6g, h). In mammalian cells, the ubiquitin-binding autophagy adaptor protein p62/SQSTM1 accumulates on dysfunctional mitochondria[52] and has been proposed to facilitate recruitment of damaged mitochondria to autophagosomes[6]. Furthermore,

recent work has shown that *Refractory to Sigma P*, *ref(2)P*, the single *Drosophila* orthologue of p62 plays a critical role in the PINK1-Parkin mitophagy pathway[53, 54]. Impaired autophagy is associated with increased levels of p62 in mammals and *Drosophila*[55], indicating that levels of p62 reflect autophagic status. To gain further insight into age-related changes in mitochondrial homeostasis, we examined *ref(2)P* co-localization with mitochondria in aging flight muscle. We observed a striking accumulation of ref(2)P co-localized with mitochondria in midlife (from day 28 to day 37), consistent with a decline in mitophagy in aged muscle (Fig. 6i, j). Importantly, short-term, midlife induction of Drp1 (day 30–37) prevented the accumulation of ref(2)P co-localized with mitochondria during aging (compare Fig. 6j, k and quantification of p62 levels in Fig. 6l). To confirm this finding, we examined ref(2)P levels specifically in the mitochondrial fraction from aged flies. Short-term, midlife Drp1 induction (day 30–37) resulted in reduced ref(2)P levels in the mitochondrial fraction from aged flies (Fig. 6m, n and Supplementary Fig. 6g). Together, these findings suggest that the midlife shift toward more elongated, less circular mitochondria, in flight muscle, contributes to a decline in mitophagy. Short-term Drp1 induction in midlife, for 7 days, promotes mitochondrial fragmentation and facilitates mitophagy. RU486 had no significant impact on mitochondrial DNA levels in aged flight muscle, co-localization of Atg8a/LC3 with mitochondria and accumulation of ref(2)P co-localized with mitochondria (Supplementary Fig. 6b, c, e, f, h and i).

**Midlife Drp1 induction requires autophagy to prolong lifespan**. To seek evidence for a causal role for autophagy in Drp1-mediated longevity, we set out to directly manipulate Atg1, a Ser/Thr protein kinase involved in the initiation of autophagosome formation[56]. More specifically, we inhibited *Atg1* by RNAi from midlife onwards in flies with increased Drp1-mediated mitochondrial fission and compared survivorship to flies with increased Drp1-mediated mitochondrial fission alone (Fig. 7a–e). We found that induced RNAi of *Atg1* in midlife reduced ATG8a/LC3 levels in aged flight muscle (Fig. 7b, d) and suppressed the lifespan extension associated with midlife Drp1 induction (Fig. 7e, Supplementary Fig. 7a). Inducing RNAi of *Atg1* in midlife did not shorten lifespan in control flies (Supplementary Fig. 7b). These results demonstrate that the lifespan-extending effects of promoting Drp1-mediated mitochondrial fission in midlife depend upon the autophagy pathway. In a similar fashion, we find that induced RNAi of *Atg1* in midlife suppressed the lifespan extension associated with midlife RNAi of *dMfn* (Supplementary Fig. 7c).

Next, we set out to determine whether autophagy is required for Drp1-mediated improvements in mitochondrial function in middle-aged flies. We found that induced RNAi of *Atg1* in midlife suppressed the increase in mitochondrial respiratory function, complex I activity and reduced ROS levels associated with midlife Drp1 induction (Fig. 7f–h). Finally, we set out determine whether autophagy was involved in Drp1-mediated improvements in proteostasis in aged animals. Importantly, we found that induced RNAi of *Atg1* in midlife suppressed the Drp1-mediated reduction in protein aggregates in aged muscle (Fig. 7i, j). Interestingly, we also observed that promoting Drp1-mediated mitochondrial fission in the context of induced RNAi of *Atg1* in midlife leads to increased levels of ubiquitinated proteins in the mitochondrial fraction from aged flies (Fig. 7k, l; Supplementary Fig. 7d).

Taken together, our data indicate that the anti-aging, prolongevity effects of promoting mitochondrial fission, in middle-aged animals, is dependent upon the activity of the autophagy pathway.

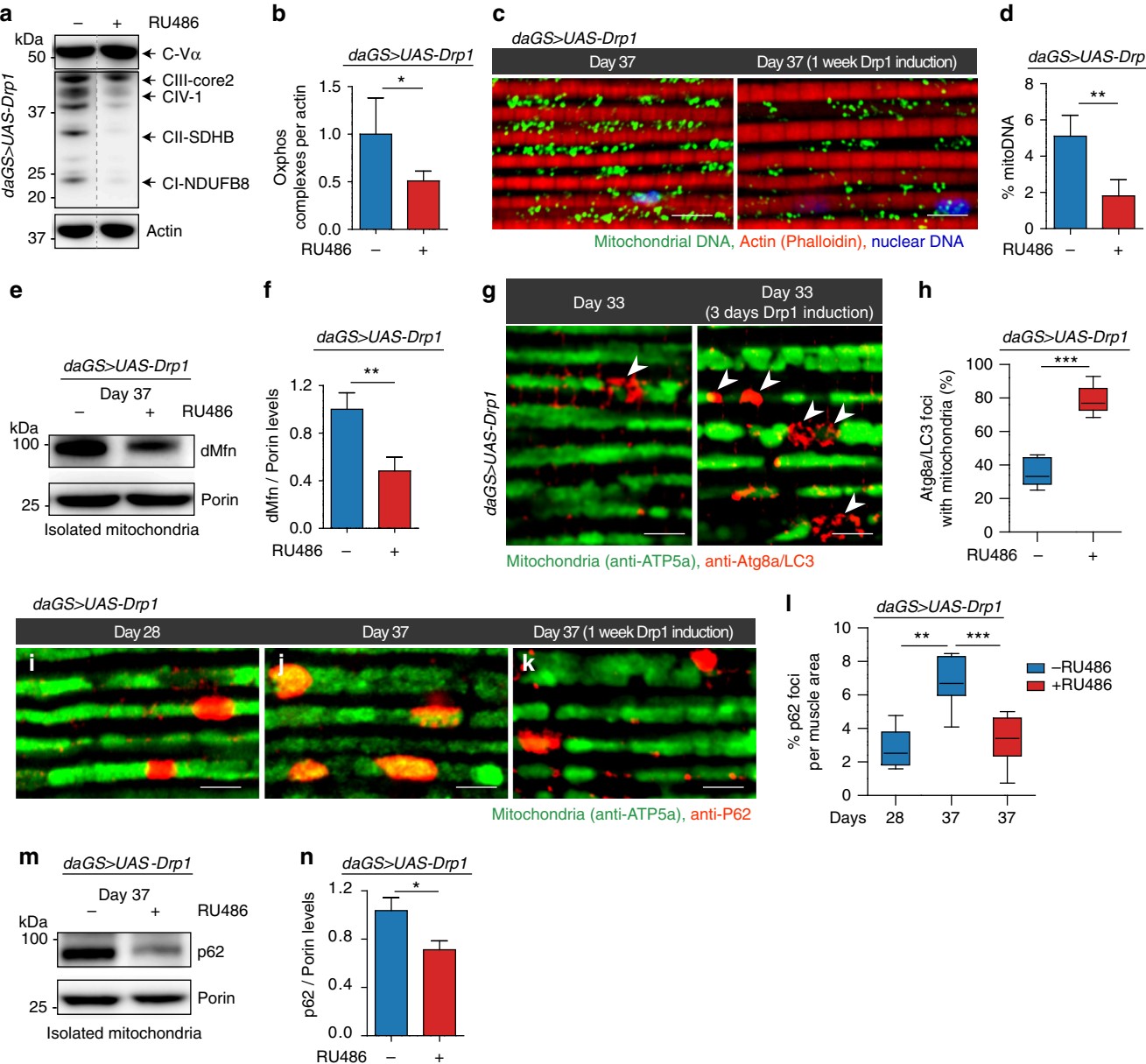

**Fig. 6** Midlife Drp1 induction facilitates mitophagy in aged flies. All experiments were carried out in *daGS > UAS-Drp1* females with or without RU486-mediated transgene induction from day 30 onwards. **a, b** Western blot **a** detection of mitochondrial respiratory complex subunits in thoraces dissected from day 37 flies. Densitometry of blots **b**; *n* = 4 replicates with 5 thoraces per replicate; *p* < 0.05; two-tailed unpaired *t*-test. **c, d** Immunostaining of indirect flight muscles **c** from day 37 flies showing mitochondrial DNA (*green channel*, anti-ds DNA antibody), nuclear DNA (*blue channel*, stained with TO-PRO-3) and muscles (*red channel*, stained with phalloidin/F-actin). Quantification of mitochondrial ds DNA **d** in muscles (as shown **c**); *n* = 4 flies; *p* < 0.05 two-tailed unpaired *t*-test. **e, f** Western blot **e** detection of mitochondrial fusion-promoting factor Mitofusin in isolated mitochondria from day 37 flies. Densitometry of blots **f**; *n* = 3 replicates with 20 flies per replicate; **p* < 0.01 two-tailed unpaired *t*-test. **g, h** Immunostaining of indirect flight muscles **g** from day 33 flies showing mitochondria (*green channel*, anti-ATP5a) and Atg8a (*red channel*, anti-Atg8a). Quantification **h** of Atg8a foci co-localizing with mitochondria (as shown in **g**); *n* = 7 flies; ***p* < 0.001; two-tailed unpaired *t*-test. **i–l** Immunostaining of indirect flight muscles from day 37 flies showing mitochondria (*green channel*, anti-ATP5a) and p62 (*red channel*, anti-p62). Quantification **l** of P62 foci per muscle area (as shown in **i–k**. *n* = 7–9 flies; ***p* < 0.01 and ****p* < 0.001; one-way ANOVA/Bonferroni's multiple comparisons test. **m, n** Western blot **m** detection of P62 levels in isolated mitochondria from day 37 flies. Densitometry of blots **n**; *n* = 3 replicates, 20 flies per replicate; *p* < 0.05; two-tailed unpaired *t*-test. Bars **b**, **d**, **f** and **n** depict mean ± s.d. Boxplots **h** and **l** display the first and third quartile, with the horizontal *bar* at the median and *whiskers* showing the most extreme data point, which is no more than 1.5 times the interquartile range from the box. *Scale bar* is 5 μm. RU486 was provided in the media at a concentration of 25 μg/ml

## Discussion

One of the major goals of geroscience research is to identify interventions that can be targeted to aged organisms to delay the onset of pathology and, thereby, prolong healthy lifespan[57]. Ideally, such interventions would not need to be applied for the long-term. In this study, we have demonstrated that short-term, midlife induction of Drp1, a Dynamin-related protein that catalyzes mitochondrial fission, can extend lifespan in both male and female *Drosophila*. Moreover, we find that short-term induction of Drp1, in middle-aged flies, delays the onset of intestinal pathology and improves additional markers of organismal health in aged flies. Reports of the impact of alterations of mitochondrial

dynamics on lifespan in different model organisms have not always been consistent. Previous studies in fungal models have reported that reducing mitochondrial fission can extend lifespan[58]. In *C. elegans*, loss of Drp1 had no impact on lifespan in wild-type worms, but rather Drp1 inactivation was shown to specifically synergize with insulin-like signaling mutants to extend lifespan[59]. Furthermore, reducing Drp1 levels has been reported to protect against certain neurodegenerative disease models[60, 61]. A major difference between these previous studies and our current study is that we have manipulated Drp1

expression in middle-aged animals. In addition, we find that RNAi of the mitochondrial fusion-promoting factor *dMfn* from midlife onwards similarly prolongs lifespan. Taken together, these findings reveal that promoting a shift in mitochondrial dynamics toward increased fission/decreased fusion in middle-aged animals prolongs organismal health and lifespan.

It has been shown previously that promoting mitochondrial fission rescues several degenerative phenotypes in PINK1 and Parkin mutant flies[62–64]. These data support the emerging model that mitochondrial fission can facilitate mitophagy[7, 11, 12, 27–29].

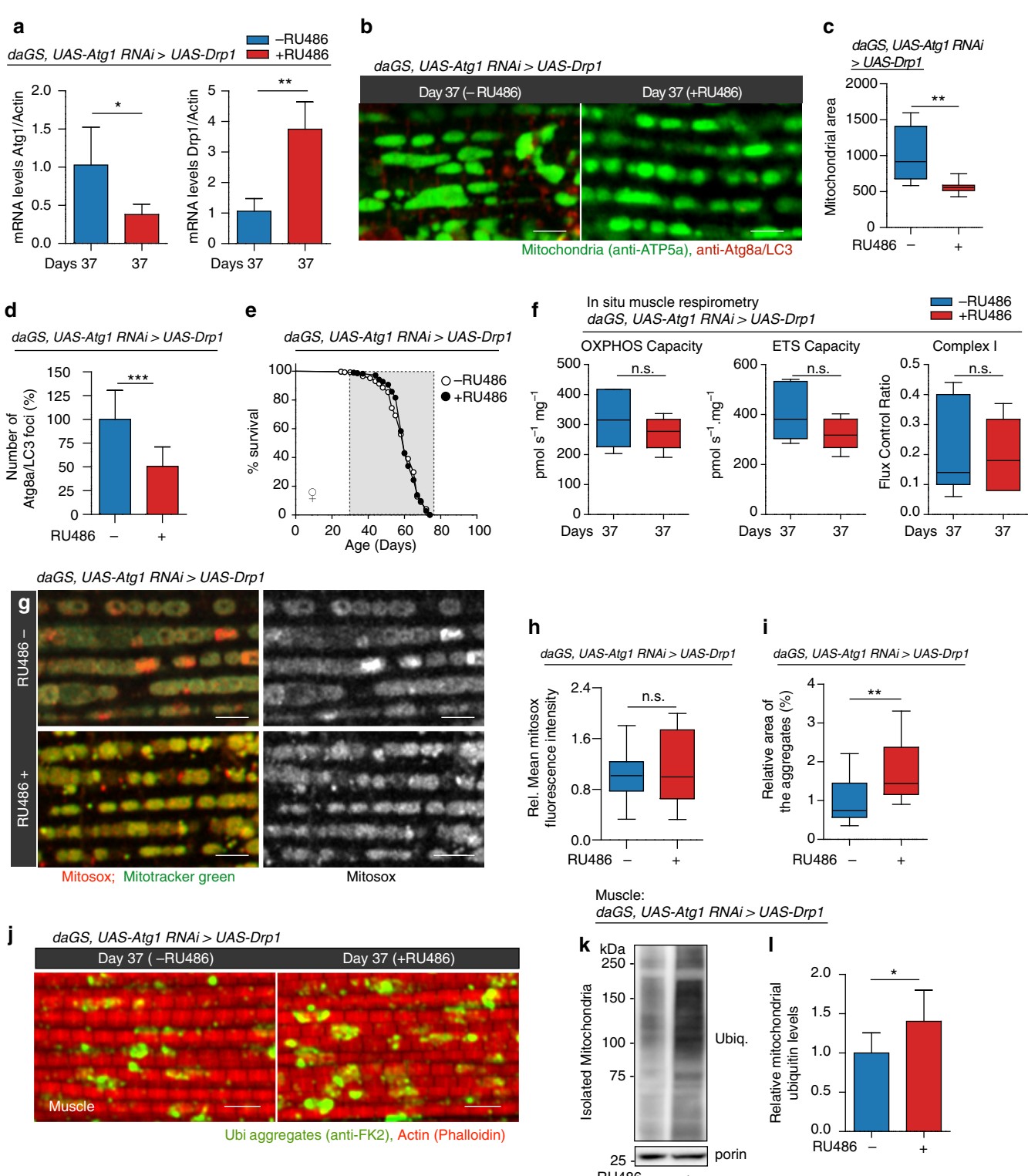

However, little was known about the relationships between mitochondrial dynamics, mitophagy, and aging. We have discovered that a midlife shift toward a more elongated mitochondrial morphology is associated with two major hallmarks of aging, mitochondrial dysfunction and loss of proteostasis, in aged flight muscle. Short-term induction of Drp1, in midlife, rejuvenates mitochondrial morphology and leads to improved mitochondrial capacities for oxidative phosphorylation and electron transport system flux. These improvements in mitochondrial function, mediated by midlife Drp1 induction, are linked to reduced mitochondrial ROS. In addition, short-term, midlife Drp1 induction restores proteostasis in aged flies. These data reveal that the accumulation of elongated dysfunctional mitochondria, in flight muscle from middle-aged animals, contributes to age-onset proteotoxicity. One potential explanation for this finding would be that a midlife decline in mitophagy, due to a shift in mitochondrial dynamics, leads to the accumulation of insoluble or aggregated mitochondrial proteins. One of the most pressing future questions to emerge from our work concerns the mechanisms that underlie the midlife shift toward a more elongated mitochondrial morphology in aged muscle. It is tempting to speculate that this may reflect a shift toward increased mitochondrial fusion, as a physiological adaption to age-onset mitochondrial damage. Indeed, fusion can mitigate the effects of localized mitochondrial damage through the exchange of proteins and lipids with other mitochondria as a form of complementation[20]. It will also be interesting to determine the precise mechanisms by which promoting mitochondrial fission in aged muscle leads to improved mitochondrial respiratory function. Consistent with a model in which Drp1-mediated longevity is linked to autophagy, we find that disrupting Atg1 suppresses the improvement in mitochondrial homeostasis, proteostasis and extended lifespan associated with midlife Drp1 induction. As Drp1 is also involved in the fission of peroxisomes[65], however, it is formally possible that altered peroxisome division may play a role in Drp1-mediated longevity. Indeed, an interesting avenue for future exploration would be to study the interplay between mitochondrial morphology, peroxisomal morphology and aging.

An additional set of questions concern the relevance of our findings to mammalian aging. Recent studies have reported a decline in mitophagy in aged mammals[15, 16] and disruptions in mitophagy have been implicated in the pathophysiology of cardiac senescence[66] and a number of neurodegenerative disorders[8, 67]. However, an understanding of the interplay between mitochondrial dynamics and mitophagy during mammalian aging is lacking. Indeed, studies of alterations in mitochondrial morphology during mammalian aging have been inconclusive. While a shift toward a more fragmented mitochondrial network has been reported in skeletal muscle from animals of advanced age[68], several studies have reported larger and less circular mitochondria in aged skeletal muscle, suggesting a shift toward hyper-fused mitochondria[69, 70]. It has been suggested that these seemingly contradictory results may be due to the fact that denervation induces mitochondrial fission in muscle[68, 71]. Therefore, studies examining muscles from very advanced age are more likely to sample regions from denervated fibers containing a highly fragmented mitochondrial morphology. Indeed, the studies finding more elongated mitochondria examined younger animals[69, 70] than the study reporting more fissioned mitochondria[68]. Moreover, reports of hyper-fused mitochondria in muscle from middle-aged animals is consistent with observations of so-called giant mitochondria in aged mammalian heart tissue[72]. Therefore, it appears that the accumulation of elongated, less circular mitochondria in midlife may also be a hallmark of mammalian aging. Critically, manipulations that cause excessive mitochondrial fusion, such as overexpression of dominant-negative Drp1[21] or nutrient starvation[73], preclude mitophagy in mammalian cells. It will, therefore, be important to determine whether promoting mitochondrial fission/inhibiting fusion in middle-aged mammals can improve mitochondrial homeostasis and delay the onset and progression of aging-related health decline.

## Methods

**Fly strains and media**. *UAS-Drp1-HA, UAS- Drp1[K38A], and UAS-Mfn* were provided by J. Chung (Korea Advanced Institute of Science and Technology, Republic of Korea). An independent *UAS-Drp1* transgene was provided by Mel Feany (Harvard Medical School). *UAS-dMfn-RNAi* and *UAS-Atg1-RNAi* lines were received from the Vienna Drosophila RNAi Center, *daughterless*-Gene-Switch (*daGS*) was provided by H. Tricoire (Université Paris Diderot–Paris7, Paris, France), and Elav–GeneSwitch (*ElavGS*) was provided by H. Keshishian (Yale University, New Haven, CT). *white[1118]* was provided by the Bloomington Stock Center. Flies were reared in vials containing cornmeal medium (1% agar, 3% yeast, 1.9% sucrose, 3.8% dextrose, 9.1% cornmeal, 1% acid mix, and 1.5% methylparaben, all concentrations given in wt/vol).

**Lifespan analysis**. Flies were collected under light nitrogen-induced anesthesia and housed at a density of 30 male or female flies per vial. All flies were kept in a humidified, temperature-controlled incubator with 12 h on/off light cycle at 25 °C. RU486 was dissolved in ethanol and administered in the media while preparing food. Flies were flipped to fresh vial every 2–3 days and scored for death.

**Starvation resistance assay**. Female flies were aged for 37 days in vials containing cornmeal medium with or without RU486 addition from 30 days. Then flies were transferred to 1% agar solution in a humidified, temperature-controlled incubator with 12 h on/off light cycle at 25 °C. Percentage of survival was measured every day, with survivors transferred to fresh vial every 2–3 days.

**Fecundity assay**. Flies were collected and kept in a humidified, temperature-controlled incubator with 12 h on/off light cycle at 25 °C. Eggs laid per fly in 24 h were counted.

---

**Fig. 7** Autophagy is required for Drp1-mediated longevity. All experiments, except **e**, were carried out on day 37 in *daGS, UAS-Atg1-RNAi > UAS-Drp1* females with or without RU486-mediated transgene induction from day 30 to 37. **a** qPCR analyses of *Atg1* and *Drp1* mRNA levels. $n = 5$ replicates (three flies per replicate); *$p < 0.05$, **$p < 0.01$; two-tailed Mann–Whitney test. **b–d** Immunostaining **b** of indirect flight muscles showing mitochondria (*green channel*, anti-ATP5a) and Atg8a foci (*red channel*, anti-Atg8a). Quantification of mitochondrial size **c** and total number of Atg8a foci **d**; $n = 6$–10 flies; **$p < 0.01$, ***$p < 0.001$; two-tailed unpaired *t*-test. **e** Survival curves with or without RU486-mediated transgene induction from day 30 onwards. The *shaded area* indicates the duration of Drp1 activation and Atg1 RNAi. $p = 0.96$, log-rank test; $n > 288$ flies. **f** In situ respirometry of permeabilized muscles to assess the capacity for oxidative phosphorylation (OXPHOS) and Electron Transport System (ETS) flux, and the flux control ratio of Complex I by rotenone inhibition. $n = 6$ replicates (two thoraces per replicate); n.s. indicates not significant; two-tailed unpaired *t*-test. **g**, **h** Staining of indirect flight muscles **g** showing mitochondria (*green channel*, Mitotracker green staining) and levels of superoxide radicals (*red channel*, staining with MitoSOX reagent). Quantification of free superoxide radicals **h**; $n = 12$ replicates; n.s. indicates not significant; two-tailed unpaired *t*-test. **i**, **j** Immunostaining of indirect flight muscles **j** showing protein polyubiquitinated aggregates (*red channel*, muscles stained with phalloidin/F-actin and *green channel*, antipolyubiquitin). **i** Quantification of polyubiquitin aggregates in muscle (as shown **j**); $n = 16$ flies; **$p < 0.01$; two-tailed unpaired *t*-test. **k**, **l** Western blot **k** detection of total ubiquitin-conjugated proteins in isolated mitochondria. Densitometry of ubiquitin blots **l** from mitochondrial pellet; $n = 8$ replicates, 25 flies per replicate; *$p < 0.05$; two-tailed unpaired *t*-test. Bars **a**, **d** and **l** depict mean ± s.d. Boxplots **c**, **f**, **h** and **i** display the first and third quartile, with the horizontal bar at the median. *Scale bar* is 5 μm. RU486 was provided in the media at a concentration of 25 μg/ml

**Analysis of food intake**. Quantification of food intake was assayed at 37 days of age using CApillary FEeding (CAFE) assay as described previously[18] with modifications. Briefly, 10 flies were placed in vials with wet tissue paper as a water source and a capillary food source [5% sucrose, 5% yeast extract, 2.5% FD&C Blue No. 1 (SPS Alfachem)]. Feeding was monitored from 11 am until 30 min after lights off (7:30 pm) in the evening, with capillaries being replaced regularly and feeding amounts recorded every 2–3 h.

**Physical activity**. Physical activity was analyzed in two different ways. *Spontaneous physical activity assay:* 10 adult female flies were placed in a *Drosophila* activity monitor (TriKinetics). Movements were recorded continuously under normal culturing conditions for 36 h on a 12-h:12-h dark:light cycle. Bar graphs represent mean activity per fly per hour for 36 h and the scatter plot shows spontaneous activity per fly during 12-h:12-h dark:light cycle. Triplicate samples were used for each activity measurement. *Climbing activity:* ~100 adult female flies were placed in 100 ml glass cylinder. Cylinder was tapped quickly and flies were allowed to settle for 2 min. This step was repeated eight times. Then cylinder was quickly tapped and after 1 min, the number of flies in upper, middle, and lower 1/3$^{rd}$ part of the cylinder were recorded.

**Quantification of triglycerides**. Lipids were extracted from three whole female flies in a chloroform:ethanol solution (2:1 vol/vol). The nonpolar lipids (fatty acid, triacylglycerol) were separated by thin-layer chromatography with n-hexane, diethylether, glacial acetic acid solution (70:30:1, vol/vol/vol). Plates were air-dried and stained with 0.2% amido black 10B in 1 M NaCl and lipid bands were quantified by photo densitometry using ImageJ and normalized to body weight.

**Intestinal barrier dysfunction (Smurf) assay**. The intestinal barrier dysfunction (Smurf) assay was performed similarly to[36]. Flies were aged on standard medium until the day of the Smurf assay. Dyed medium was prepared using standard medium with F&D blue dye #1 (purchased from SPS Alfachem) added at a concentration of 2.5% wt/vol. Flies were transferred onto dyed medium for 16 h and a fly was counted as a Smurf when dye coloration was observed outside the digestive tract.

**Antisera generation**. Rabbit polyclonal antiserum recognizing *Drosophila* Atg8a and refractory to Sigma P, ref(2)P, the single *Drosophila* orthologue of p62 were generated by a commercial source (AbMart Inc., China) using synthetic peptides corresponding to sequences in these proteins (Atg8a: MKFQYKEEHAFEKRR; p62: VNTDQSVPRTED). The serum was antigen-affinity-purified and specificity of the antibody was shown by immunostaining (Supplementary Fig. 8a, b).

**Immunostaining and image analysis**. Flies were fixed for 20 min with 3.7% formaldehyde in PBS and hemi-thoraces were dissected. For brain immunostaining, brains were dissected and fixed for 20 min with 3.7% formaldehyde in PBS. Samples were then rinsed three times with 0.2% Triton X-100 in PBS (PBST) and blocked for 1 h at room temperature in 3% BSA in PBST. Primary antibodies were diluted 1:250 in 5% BSA in PBST and incubated overnight at 4 °C. Primary antibodies used were: rabbit anti-Drp1 (a generous gift from Dr. Leo Pallanck[11]), mouse anti-ATP5A (15H4C4, Abcam), anti-ds DNA (ab27156, Abcam), anti-HA (HA.11 Clone 16B12 Monoclonal Antibody, Covance), Mono- and poly-ubiquitinylated conjugates monoclonal antibody (FK2) (BML-PW8810, Enzo), rabbit anti-ATG8a (see antisera generation) and p62 (see antisera generation). Samples were then rinsed thrice with PBST and incubated with secondary antibodies and/or stains for 3 h at room temperature. Secondary antibodies and stains were diluted 1:250 in 5% BSA in PBST and incubated for 3 h at room temperature. Secondary antibodies used were: anti-rabbit or anti-mouse AlexaFluor-488 (Invitrogen); anti-rabbit or anti-mouse AlexaFluor-555 (Invitrogen); To-Pro-3 DNA stain (Invitrogen); phalloidin AlexaFlour-568 (Invitrogen). Samples were then rinsed three times and mounted in Vectashield mounting medium (Vector Labs) and imaged using Zeiss single point LSM 5 exciter confocal microscope. Mitochondrial or protein aggregates area measurements were done using ImageJ. Analyze particle feature of ImageJ was used to measure the area of mitochondria or the protein aggregate. For protein aggregates, this value was corrected for the area of muscle and represented as % protein aggregates per defined muscle area.

**Electron microscopy**. Standard procedure for electron microscopic (EM) analysis was carried out as described in ref. [74] with slight modifications. Thoraces were fixed in 2% glutaraldehyde and 2% formaldehyde in 0.1 M sodium phosphate buffer (PB) containing 0.9% NaCl overnight at 4 °C. Thoraces were then bisected and post-fixed in 1% osmium tetroxide in PB, treated with 0.5% uranyl acetate, and dehydrated through a graded series of ethanol concentrations. After infiltration with Eponate 12 resin, the samples were embedded in fresh Eponate and polymerized overnight. Semi-thin sections (1.5 µm) were cut on an ultramicrotome and stained with toluidine blue. The muscle area of interest was identified from these sections. For EM, sections of 50 nm thickness were prepared from the identified area, placed on formvar coated copper grids and stained with uranyl acetate and Reynolds' lead citrate. The grids were examined using a JEOL 100CX transmission electron microscope at 60 kV(Electron Microscopy Facility, UCLA Brain Research Institute). For quantification, ImageJ was used to measure mitochondrial perimeter from electron micrographs taken under identical conditions and magnification.

**TMRE staining**. Flies were anesthetized and hemi-thoraces were dissected in cold *Drosophila* Schneider's Medium (DSM) (Thermo Fisher Scientific). Hemi-thoraces were then incubated in TMRE staining solution, consisting of 100 nM of TMRE (Thermo Fisher Scientific) in DSM, for 12 min at room temperature. Samples were rinsed two times for 30 s each wash with a solution consisting of 25 nM of TMRE in DSM. Samples were quickly mounted in the same medium onto the slide and imaged within 15–20 min using identical setting on the confocal microscope. Quantification of TMRE staining is done using Image J in which mean intensity values for the TMRE stains were quantified.

**Mitochondria purification**. Samples were gently crushed in chilled mitochondrial isolation medium (MIM) [250 Mm sucrose, 10 mM Tris-HCl (pH 7.4), 0.15 mM MgCl$_2$] using a plastic pestle homogenizer and then spun twice at 500 × g for 5 min at 4 °C to remove debris. The supernatant was then spun at 5000 × g, for 5 min at 4 °C. The pellet, containing the mitochondria, was stored at −80° C.

**Complex I assay from mitochondrial pellet**. Complex I assay was performed as described in ref. [18]. The mitochondrial pellet was washed with MIM and resuspended in 100 µl of MIM or assay buffers. Dilutions of mitochondrial preparations were resuspended in NADH dehydrogenase assay buffer (1 × PBS, 0.35% BSA, 200 µM NADH, 240 µM KCN, 60 µM 2,6-dichloroindophenol sodium salt hydrate (DCIP), 70 µM decylubiquinone, 25 µM antimycin A) containing 2 µM rotenone or ethanol. Complex I activity was measured as the rate of decrease in absorbance at 600 nm using an Epoch plate-reading spectrophotometer (BioTek).

**In situ muscle and head respirometry**. Mitochondrial respiration was measured in isolated thoraces or heads in situ at 25 °C in hyperoxygenated media (550–350 nmol/ml) using high-resolution respirometry (O2k, Oroboros, AT). After rapid dissection, whole thoraces were permeabilized in saponin (50 µg/ml) prepared in ice-cold BIOPS for 15 min (BIOPS: 2.77 mM CaK$_2$ EGTA, 7.23 mM K$_2$ EGTA, 5.77 mM Na$_2$ ATP, 6.56 mM MgCl$_2$·6H$_2$O, 20 mM taurine, 15 mM Na$_2$ PCr, 20 mM imidazole, 0.5 mM DTT, 50 mM MES hydrate), followed by 2 × 5 min washes in respiration medium (MiR05: 0.5 mM EGTA, 3 mM MgCl$_2$, 60 mM K-lactobionate, 20 mM taurine, 10 mM KH$_2$PO$_4$, 20 mM HEPES, 110 mM sucrose, and 1 g/l BSA, pH 7.1). After washing, thoraces were gently dried on filter paper and weighed before being placed into the respirometer chambers (2 flies/chamber). Similarly for head respirometry, 10 heads were dissected in saponin (25 µg/ml) prepared in ice-cold BIOPS washed quickly and placed inside the respirometer chamber. Oxidative phosphorylation (OXPHOS) and electron transport system (ETS) capacities were measured using a modified substrate-uncoupler-inhibitor-titration protocol that consisted of multiple sequential injections at saturating concentrations [Bratic, A. et al, 2015 and Pichaud, N. et al, 2011]. To achieve maximal ADP-stimulated respiration from electron flux through complex I, 5 mM pyruvate, 5 mM proline, 2 mM malate, 10 mM glutamate, and 2.5 mM ADP were added to the chambers for thoraces and 5 mM pyruvate, 5 mM proline, 2 mM malate, and 2.5 mM ADP were added for the heads. To achieve maximal convergent electron flux through both complex I and II (i.e., OXPHOS capacity), 15 mM glycerol-3-phosphate and 10 mM succinate were injected into each chamber. To assess the functional integrity of the outer mitochondrial membrane, 10 µM cytochrome c was added. Samples that increased OXPHOS by >15% during this step were deemed to have poor integrity and rejected on the basis of quality control [Kuznetsov, A.V. et al., 2008]. Non-phosphorylating LEAK respiration in the presence of high adenylates (L$_{Omy}$) was evaluated by inhibiting ATP synthase with 2.5 µM oligomycin. To assess electron transport system (ETS) capacity, 0.5 µM carbonylcyanide p-trifluoromethoxy-phenylhydrazone (FCCP) was titrated in 1 µl steps. 0.5 µM rotenone was then added to inhibit complex I and calculate the complex I contribution to ETS capacity. Lastly, residual oxygen consumption (non-mitochondrial respiration) was determined by adding 2.5 µM Antimycin A to inhibit complex III. The total assay time per sample was ~1.5 h. Oxygen flux for each respiratory state was adjusted for sample weight and corrected by subtracting the non-mitochondrial respiration. The flux control ratio for complex I was calculated from the rate of oxidation after rotenone injection expressed as a fraction of ETS capacity (after FCCP steps), and subtracted from one.

**MitoSox staining assay**. Flies were anesthetized and hemi-thoraces were dissected in cold *Drosophila* Schneider's Medium (DSM) (Thermo Fisher Scientific). Hemi-thoraces were then incubated in a staining solution consisting of 5 µM MitoSOX Red (Thermo Fisher Scientific) and 100 nM MitoTracker Green (Thermo Fisher Scientific) in DSM. Staining was done for 12 min at room temperature and then samples were rinsed two times for 30 s each wash with DSM. Samples were quickly mounted in DSM and imaged within 10–15 min using identical confocal microscope settings. Quantification of staining is done using Image J in which mean intensity value for the MitoSOX stain was quantified.

**Quantitative real-time PCR**. Total RNA was extracted using TRIzol reagent (Invitrogen) following manufacturer protocols. Samples were treated with DNAse, and then cDNA synthesis was carried out using the First Strand cDNA Synthesis Kit from Fermentas. PCR was performed with Power SYBR Green master mix (Applied Biosystems) on a BioRad Real-Time PCR system. Cycling conditions were as follows: 95 °C for10 min; 95 °C for 15 s then 60 °C for 60 s, cycled 40 times. Equalized amplicons of Actin5C were used as a reference to normalize. Primer sequences were as follows:

Act5C, TTGTCTGGGCAAGAGGATCAG and ACCACTCGCACTTGCACT TTC;

Drp1, ATTGTTGTTCTAGGCAGTCAG and GAACTCTTGCCGGAGCT Hsp60, TGATGCTGATCTCGTCAAGC and TACTCGGAGGTGGTGTCCTC; Hsp10, CCCGCATCTAGCGAGAATAG and CTCCTTTCGTCTTGGTCAGC; Hsc70-5, GGAATTGATATCCGCAAGGA and TCAGCTTCAGGTTCATG TGC;

Atg1, CGTCTACAAAGGACGTCATCGCAAGAAAC and CGCCAAGTCGC CGCCATTGCAATACTC.

**Western blot assay**. Samples were collected and lysates were separated by SDS page using standard procedures. Membranes were probed with antisera against anti-actin (PA5-16914, Thermo Fisher Scientific), anti-HA (HA.11 Clone 16B12 monoclonal Antibody, Covance), Anti-VDAC1/Porin (ab14734, Abcam), anti-p62 (see antisera generation), anti-dMfn (a generous gift from Dr. Leo Pallanck), MitoProfile Total OXPHOS Rodent WB Antibody Cocktail (ab110413, Abcam) and anti-Ubiquitin (P4D1, mouse mAb no. 3936 from Cell Signaling). All primary antibodies were used in 1:2500 dilutions except anti-actin where dilution was 1:15,000. The rabbit antibodies were detected using horseradish peroxidase-conjugated anti-rabbit IgG antibodies (1:2000 dilution; Sigma). The mouse anti-bodies were detected using horseradish peroxidase-conjugated anti-mouse IgG antibodies (1:2000 dilution; Sigma). Amersham ECL Prime Western Blotting Detection Reagent (GE life sciences) was used to visualize the presence of horse-radish peroxidase, and the chemiluminescent signal was recorded using Syngene Pxi Western Blot Imager. Image analysis was done using ImageJ.

**Statistical analysis**. Prism5 (GraphPad) was used to perform the statistical analysis and graphical display of the data. Significance is expressed as p values which was determined with two-tailed, unpaired, parametric or non-parametric tests as indicated in the figure legends unless otherwise stated in material and methods. For two group comparisons: either unpaired t-test was used when data met criteria for parametric analysis (normal distribution assessed by Shapiro-Wilk normality test or Kolmogorov–Smirnov test using Prism5 and equal variance assessed by Fisher's exact test using excel) or Mann–Whitney U-test was used in case of non-parametric analysis. For more than two group's comparison ANOVA with Bonferroni post hoc test was performed. Kruskal–Wallis with dunn's post hoc test was used when data did not meet requirements for parametric test. For comparison of survival curves, Log-rank (Mantel-Cox) test was used.

**Data availability**. The data that support the findings of this study are available from the corresponding author upon reasonable request.

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

## Acknowledgements

We thank M. Feaney, L. Pallanck, L. Jones, Vienna Drosophila RNAi Center, and the *Drosophila* Stock Center (Bloomington) for fly stocks; L. Pallanck for antibodies; M. Cilluffo for help with electron microscopy; and the L. Jones, M. Frye, and D. Simmons labs for use of their equipment. M.P.O. was supported by a CAPES scholarship (grant 10053/14-0). A.V.K. was supported by a fellowship from The Pulmonary Education and Research Foundation. This work was supported by NIH grants (R01AG037514, R01AG049157, and R01AG040288) to D.W.W. Stocks obtained from the Bloomington Drosophila Stock Center (NIH P40OD018537) were used in this study. This research was conducted while D.W.W. was a Julie Martin Mid-Career Awardee in Aging Research supported by The Ellison Medical Foundation and AFAR.

## Author contributions

A.R. and D.W.W. designed the experiments. M.R. conducted Drp1 lifespan experiments. M.P.O. conducted healthspan assays and respirometry experiments. A.V.K. conducted respirometry experiments in H.B.R.'s lab and analyzed the data. R.A. helped conduct immunohistochemistry experiments. A.R. conducted all other experiments and analyzed the data. A.R. and D.W.W. wrote the manuscript.

## Additional information

**Competing interests:** The authors declare no competing financial interests.

