## [Peer Review File · Nature Communications]

Reviewers' Comments:

Reviewer #1 (Remarks to the Author)

In this manuscript, Walker and colleagues showed that short-term ectopic expression of the mitochondrial fission regulator Drp1 is sufficient to prolong healthspan and lifespan in *Drosophila*. Their genetic interaction studies indicated that the effect of Drp1 overexpression on lifespan is ATG1-dependent, supporting the notion that mitophagy is involved. This work is a logical extension of a previous study from the same lab showing that overexpression of Parkin, a key regulator of mitophagy, also extends *Drosophila* lifespan. In light of the similar theme of the two studies, it appears that the current study lacks sufficient mechanistic insight to advance what is already known from the previous study.

Main comments:

Is there age-dependent decline of Drp1 expression or activity? Does Drp1 down-regulation midlife cause shortening of lifespan??

How does mitochondrial morphology/function affect proteostasis? what is the molecular link?

Most of the mitochondrial analyses are focused in flight muscle, but the anti-aging effect of Drp1 manipulation occurs in neurons or intestine. It is therefore important to show that what is observed in flight muscle applies to neurons or intestinal cells.

To further test the idea that midlife Drp1 induction promotes lifespan by promoting mitophagy, the experiment should be performed in PINK1 or Parkin mutant condition, where mitophagy is blocked.

Minor comments:

Is the pro-longevity effect of Marf RNAi also ATG dependent?

Reviewer #2 (Remarks to the Author)

Rana and col. investigate a highly relevant topic: the role of mitochondrial dynamics in the rate of ageing. The authors describe that old flies have more elongated (hyper-fused) mitochondria, which they claim correlates with a reduction in mitochondrial function. Next, they show that up-regulation of Drp1 (a protein essential for mitochondrial fission) extends lifespan when it is overexpressed in midlife. Interestingly, short-term induction of Drp1 (7 days) elicits the same effect. Finally, the authors claim that the lifespan extension reported requires the autophagy machinery of the cell.

I find the manuscript very interesting and well written, however some few experiments are required to back up the main conclusions of the work and some controls are missing. It seems to me that there are some mistakes in the way results are represented in the figures (see below).

Major comments

(1) There is one relevant lifespan experiment missing. What are the consequences of overexpressing Drp1 continuously during lifespan? Since there is no negative effect when Drp1 is

expressed early in life, I predict that continuous expression should extend lifespan. If that is not the case, experiments to check the effect of continuous Drp1 expression on mitochondrial function in old flies (37 days) need to be produced.

(2) The data shown suggest that hyper-fusion of mitochondria in old age is deleterious and contribute to ageing and age-associated deleterious changes. However, this somehow contradicts papers showing that mitochondrial fusion is a mechanism of protection against mitochondrial damaged. To clarify the role of mitochondrial fusion in the context of ageing, the authors can overexpress pro-fusion proteins such as Marf or Opa1 in young or middle age flies and see which is the effect on lifespan. If fusion is deleterious lifespan will be shorten.

(3) Which is the effect of specific overexpression of Drp1 in muscle? Since most of the data about mitochondrial function have been performed in muscle, this experiment should be included. If lifespan is not extended assays of mitochondrial function in gut and/or brain should be performed.

(4) Is the effect of Drp1 mediated through canonical mitophagy via Pink1/Parkin? If the authors have data of the effects on lifespan of overexpressing Drp1 on a background where Pink1 and Parkin are depleted, I would strongly recommend them to include these data in the manuscript. However, this experiment is not absolutely necessary to support the main conclusions of the paper.

(5) Some controls are missing:

a. Figure 4a-d. Young controls are required to show that CI activity and mitochondrial respiration are decreased and ROS are increased in old individuals, and that these phenotypes are rescued by Drp1 expression. Why did the authors choose 44d for figure b? They don't use this age in any other experiment and don't show data of mitochondrial fission/fusion levels at this age.

b. Figure 5f-j. Similarly, the authors should show young controls.

c. Figure S3C-E young controls are missing.

d. Figures S4a-c young controls are missing.

(6) Figure 7. The authors show that Drp1 overexpression rescues the elongation phenotype associated with age in an Atg1 depleted background, but lifespan is not extended. Is it mitochondrial respiration and complex I activity increased? Are ROS reduced? These experiments should be performed to demonstrate that mitochondrial turnover is not working when Atg1 is knock-down.

(7) How many times have been the lifespan experiments repeated? Since differences are small in some experiments, lifespan studies should have at least 2-3 repetitions. A summary of the lifespan data can be presented as in a former paper from the same group: Ulgherait et al. Cell Reports. 2014.

(8) Data are presented as mean \pm SEM in most figures, however some of the figures do not represent the mean or SEM e.g. 1b or 2f. Please check this.

Minor comments:

(1) Are blots represented in figures 6a and S6a the same? Only one blot should be shown (better S6a). In addition, quantification in 6b should be done for each specific protein.

(2) Figure 7. Protein or mRNA levels of Atg1 should be shown to show that the knock-down is working.

(3) The authors should cite those other papers that show different results than the ones observed by them, for example the work of the Osiewacz's lab in yeast, several papers in *C. elegans* (e.g. Regmi and col. 2014. Aging and Yang and col. 2011. Aging Cell). These papers do not contradict their work, but indicate different mechanisms working in different species.

Reviewer #3 (Remarks to the Author)

The manuscript entitled "Promoting Drp1-mediated mitochondrial fission in midlife prolongs healthy lifespan" from Rana A. et al. describes that transient induction of the dynamin-related protein Drp1 in midlife of fruit flies decelerates their aging and increases their fitness. The data shown in the manuscript are very robust, and the experiments are well-done. The findings are

unexpected, when comparing with the findings from other model systems where Drp1 plays a detrimental role (see below). The manuscript proposes a paradigm of aging with a protective role of mitochondrial fragmentation. Due to its potential relevance for mammalian systems, the topic of this manuscript is of very high interest for the general readership of Nature Communications. However, before being considered as manuscript to be published in this journal, the authors have to overcome some major concerns and deal with minor improvements as listed below.

Major concerns:

1.) The reader of the manuscript may get the impression that it is to be expected that mitochondrial fragmentation is beneficial for cells, i.e. increased health and increased lifespan. But this is not true. Deletion of the yeast and worm homologues of Drp1 resulted in increased lifespans, and not in decreased lifespans (e.g., Scheckhuber CQ et al., *Nat Cell Biol* 2007, PMID: 17173038; Yang CC et al., *Aging Cell* 2011, PMID: 21463460). Similarly, inhibiting Drp1 expression may be protective in some neurodegenerative scenarios (e.g., Manczak M et al., *Hum Mol Genet* 2016, PMID: 27677309; Kandimalla R et al., *Hum Mol Genet* 2016, PMID: 27634647). In contrast, in mammalian cells in vitro and in vivo knock-out of Drp1 resulted in severe loss of survival leading to neurodegeneration (e.g., Kageyama Y et al., *J Cell Biol* 2012). This could be the result of the accumulation of damaged mitochondria upon defective mitophagy, whose efficiency depends (in some model systems) on mitochondrial fragmentation. The findings of the authors favor the second concept (which is therefore not very novel). However, the authors have to mention and discuss the discrepancies, which can be found in the literature. Similarly, it has to be noted that mitochondrial fragmentation is not in all cases required for mitophagy, and that mitochondrial fragmentation does not in all cases trigger mitophagy. One might get the impression when reading the manuscript that these things are obvious. Having the discrepancies of protective/detrimental roles of Drp1 in mind, I strongly recommend further substantiating the protective role of Drp1 for health and lifespan in fruit flies. I recommend expressing dominant-negative Drp1 mutants in fruit flies at the very same conditions as wild-type Drp1 (at least for one or two key experiments). If the model of the authors is true, life span and health should decline when expressing dominant-negative Drp1 mutants, whereas expressing wild-type Drp1 is protective. Alternatively, the authors may perform a knock-down of Drp1.

2.) Drp1 triggers mitochondrial fragmentation, but it is also critically involved in the division of peroxisomes. From my point of view, it cannot be excluded that increased peroxisomal fission is the protective aspect in the scenarios describes by the authors. Promoting peroxisomal fission might lead to reduced ROS levels, thereby reducing the damage of mitochondria, increasing health and lifespan. I've noticed that silencing the mitochondrial fusion factor dMfn1 had similar protective effects than expressing Drp1. Therefore, the mitochondrion-based scenario is likely, and no additional experiments are needed, but the authors should at least write that a role of peroxisomes on health- and lifespan prolongation cannot be excluded.

3.) The finding that Drp1 induction in midlife reduces the levels of ubiquitylated aggregates, which co-migrate with mitochondrial fractions, is very fascinating (for me the most fascinating aspect of the paper at all). This raises the question of the protein species, which are ubiquitylated. I could imagine that these are mitochondrial precursor proteins, which get ubiquitylated, if they cannot be imported properly into mitochondria. Mitochondrial protein import depends on mitochondrial membrane potential, which is obviously markedly reduced in fruit flies upon aging (see Fig. 3k). Thus, Drp1-mediated mitochondrial fission may restore the mitochondrial import machinery, preventing the detrimental accumulation of mitochondrial precursor protein in the cytosol. The aberrant accumulation of mitochondrial precursor proteins has been reported very recently in different model systems (Wang X et al., *Nature* 2016, PMID: 26192197; Wrobel L et al., *Nature* 2016, PMID: 26245374). I strongly recommend enriching ubiquitylated proteins from mitochondrial fractions, and sending them to MS for identification. If you see the enrichment of mitochondrial precursor proteins in the cytosol upon aging, which is reduced when Drp1 is expressed, this would favor the model as described above.

3.) How do you explain that Drp1 expression reduces the levels of some respiratory chain complexes (see Fig. 6a) and the finding that Drp1 expression increases ETS and OXPHOS activities (see Fig. 4)? I am puzzled about these counterintuitive results... I've realized that you have discussed that some respiratory chain complexes might be selectively targeted to mitophagy, but why should this enable increase respiratory chain activities?

4.) The authors have shown markers of mitophagy upon expression of Drp1 (see Fig. 6), and suggest that increased mitophagy upon Drp1-dependent mitochondrial fragmentation is beneficial, because it reduces the number of damaged mitochondria. I am not yet convinced whether this is true. The increased recruitment of Atg8a/LC3 to mitochondria (see Fig. 6g+h) could be a result of general autophagy induction upon Drp1 expression, which leads to more Atg8a/LC3 dots in general that therefore co-localize more often with mitochondria. I am not familiar with mitophagy in fruit flies, but if there is a mitophagy-specific factor that upon knock-down interferes with autophagy, the authors should knock-down this factor (interfering with mitophagy), in order to prove that mitophagy is the beneficial pathway required for the protective effect of Drp1 expression. If this is not possible, the authors should at least state that their data suggest that mitophagy is the protective effect, they shouldn't write that their data demonstrate or indicate a crucial role of mitophagy. This statement is too strong.

5.) The authors demonstrate that general autophagy is needed for the protective effect of Drp1 (see Fig. 7). I am wondering why this is the case. The authors suggest that mitophagy is pivotal here, which might be the case (see 4.). But I think that induction of general autophagy could also reduce the levels of (mistargeted) ubiquitylated (mitochondrial) proteins in the cytosol (see Fig. 5). I strongly recommend analyzing the levels of ubiquitylated proteins in cells expressing Drp1 but lacking Atg1. Drp1 (similar to Fig. 5). If Drp1 triggers autophagy leading to a decrease of damaged mitochondria, and in parallel to a decrease in ubiquitylated proteins in the cytosol, this would be very fascinating.

6.) The information about the statistical tests used in this manuscript is insufficient. A paragraph dealing with this information must be included in the Material & Methods section. The Student's t-test (unpaired or paired? One-tailed or two-tailed?) used in most cases is only adequate if comparing two groups of data. Have you used suitable pre-tests, such as equal variance and normality tests, in order to test whether you are allowed to use the Student's t-test at all? If you have three or more groups as depicted your graphs other statistical tests, such as ANOVA are needed, and the pairwise comparisons should then be performed afterwards using suitable posthoc tests (Holm-Sidak, Tukey or equivalents). Similar to Student's tests, suitable pre-tests must be performed to test whether ANOVA can be used at all. Please define for every experiment what you mean with "n": The number of independent experiments? The number of biological replicates within one or several independent experiments? The number of flies?

Minor points:

1.) Introduction (page 3, lines 61-62): "mitochondrial outer membrane proteins" instead of "outer mitochondrial membrane proteins" (for consistency)

2.) Discussion (page 11, lines 239-240): Drp1 induction correlates with decreased dMfn levels, it is not clear whether there is a causative connection between the observations.

3.) Discussion (page 12, lines 262-264): These findings SUGGEST, they do not indicate (that's too strong)

4.) Discussion (page 12): Please refer to Fig. 7b

5.) Supplementary data (pages 10/11): "radicals" instead of "radicles"

We would like to thank the reviewers for their helpful and constructive comments and suggestions to help improve the manuscript. As you will see below, we have responded to each and every concern, suggestion or comment that the reviewers raised. Indeed, we now include a significant amount of new data to further support our findings. We hope that the reviewers will agree that the manuscript has been improved as a result.

As outlined in the decision letter, we have addressed the following issues:

- 1) *Please investigate whether Drp1 might affect lifespan via mechanisms that do not depend on mito fragmentation (eg by overexpressing dominant-negative DRP1 mutants, as suggested by R3).*

We now include data showing the impact of midlife overexpression of dominant-negative Drp1 mutants on lifespan (as suggested by R3). To be clear, however, the rationale for this requested experiment was to “*further substantiate the protective role of Drp1 for health and lifespan in fruit flies*”:

R3: *“I strongly recommend further substantiating the protective role of Drp1 for health and lifespan in fruit flies. I recommend expressing dominant-negative Drp1 mutants in fruit flies at the very same conditions as wild-type Drp1 (at least for one or two key experiments). If the model of the authors is true, life span and health should decline when expressing dominant-negative Drp1 mutants, whereas expressing wild-type Drp1 is protective.”* **Importantly, we now show that midlife expression of dominant-negative Drp1 shortens lifespan and impairs health.**

- 2) *Please confirm key findings shown in muscle tissue in neurons/intestine (see comments from reviewer #1 and #2)*

This revised manuscript now includes data from brain tissue showing that midlife Drp1 induction induces mitochondrial fragmentation, improves mitochondrial OXPHOS activity, ETS capacity, complex I activity and also improves proteostasis

- 3) *We ask that you strengthen the evidence that lifespan extension mediated by transient Drp1 OE requires specifically mitophagy, as opposed to a more general induction autophagy (eg by performing experiments in flies lacking either Pink/Parkin or Atg1, as suggested by R2+3)*

Performing the experiments in flies lacking either Pink/Parkin would not lead to data that could be interpreted in a meaningful manner. In our point-by-point response to R1, we elaborate upon this point extensively. Critically, neither R2 nor R3 considered this experiment to be a requirement for publication.

R2: *“If the authors have data of the effects on lifespan of overexpressing Drp1 on a background where Pink1 and Parkin are depleted, I would strongly recommend them to include these data in the manuscript. However, this experiment is not absolutely necessary to support the main conclusions of the paper.”*

R3: "if there is a mitophagy-specific factorif this is not possible, the authors should at least state that their data suggest that mitophagy is the protective effect, they shouldn't write that their data demonstrate or indicate to a crucial role of mitophagy.

Regarding Atg1, R2 made the excellent suggestion to determine whether Atg1 was required for Drp1-mediated improvements in mitochondrial respiratory activity, complex I activity, mitochondrial ROS levels. We have now carried out these experiments. This new data strongly supports the model as pointed out by R2 that "mitochondrial turnover is not working when Atg1 is knocked-down."

4) We encourage you to strengthen the mechanistic links between morphological changes in mitochondria and proteostasis (see comments from reviewer #1 and #3).

We now include a significant amount of new data that strengthens the mechanistic links between alterations in mitochondrial morphology and proteostasis in aging flies. We believe that these new data are timely and exciting. In this revised manuscript, we now include data to show that expression of a dominant-negative Drp1 (Drp1-DN) transgene in midlife impairs proteostasis (Fig. 5E-H). Thus, strengthening our previous data showing that midlife Drp1-mediated mitochondrial fission improves proteostasis in aged flies. Furthermore, we now show that up-regulating Mfn (promoting fusion) in midlife impairs proteostasis (Fig. S5C). Taken together, our data supports a model in which a midlife shift towards a more elongated mitochondrial morphology in flight muscle impairs the clearance of damaged mitochondria/mitochondrial proteins via mitophagy. We show that promoting mitochondrial fission in midlife facilitates mitophagy and/or promotes the turnover of some mitochondrial proteins (Fig. 6). We hypothesize that facilitating mitophagy (by promoting fission) in midlife reduces the levels of damaged/insoluble (mitochondrial) proteins in aged muscle tissue. Indeed, it has been reported that there is an over-representation of mitochondrial proteins in the insoluble protein fraction from aged animals^{1,2}.

Furthermore, we now show that Atg1 is required for Drp1-mediated improvements in proteostasis in middle-aged animals. Critically, we now include data showing that promoting Drp1-mediated mitochondrial fission in Atg1 depleted animals leads to increased levels of ubiquitinated proteins in the mitochondrial fraction from aged muscle tissue.

5) Please also address all the more technical points raised by all reviewers, and improve n numbers and statistics, where needed

We have now done so.

Please find our point-by-point response to the reviewers below:

Reviewer #1 (Remarks to the Author):

In this manuscript, Walker and colleagues showed that short-term ectopic expression of the mitochondrial fission regulator Drp1 is sufficient to prolong healthspan and lifespan in Drosophila. Their genetic interaction studies indicated that the effect of Drp1 overexpression on lifespan is ATG1-dependent, supporting the notion that mitophagy is involved. This work is a logical extension of a previous study from the same lab showing that overexpression of Parkin, a key regulator of mitophagy, also extends Drosophila lifespan. In light of the similar theme of the two studies, it appears that the current study lacks sufficient mechanistic insight to advance what is already known from the previous study.

Main comments:

Is there age-dependent decline of Drp1 expression or activity? Does Drp1 down-regulation midlife cause shortening of lifespan??

Yes, we now show that there is an age-related decline in Drp1 mRNA levels (Fig. S3C). Thanks for this suggestion. This new data strengthens our finding that in midlife there is a shift in mitochondrial dynamics towards reduced mitochondrial fission.

Yes, we now show that expression of a dominant-negative Drp1 (Drp1-DN) transgene in midlife shortens lifespan (Fig. S1h). Additionally, we now include new data showing that promoting mitochondrial fusion (via Mfn up-regulation) in midlife shortens lifespan (Fig. S1i).

How does mitochondrial morphology/function affect proteostasis? what is the molecular link?

To provide further insight into the molecular link between mitochondrial morphology/function and proteostasis, we have carried out a significant amount of new experimental work. In this revised manuscript, we now include data to show that expression of a dominant-negative Drp1 (Drp1-DN) transgene in midlife impairs proteostasis (Fig. 5E-H). Thus, strengthening our previous data showing that midlife Drp1-mediated mitochondrial fission improves proteostasis in aged flies. Furthermore, we now show that up-regulating Mfn in midlife impairs proteostasis (Fig. S5C). Taken together, our data supports a model in which a midlife shift towards a more elongated mitochondrial morphology in flight muscle impairs the clearance of damaged mitochondria/mitochondrial proteins via mitophagy. We show that promoting mitochondrial fission in midlife facilitates mitophagy and/or promotes the turnover of some mitochondrial proteins (Fig. 6). We hypothesize that facilitating mitophagy (by promoting fission) in midlife reduces the levels of damaged/insoluble (mitochondrial) proteins in aged muscle tissue. Indeed, it has been reported that there is an over-representation of mitochondrial proteins in the insoluble protein fraction from aged animals^{1,2}.

Furthermore, we have further expanded upon our previous data to now show that the ability of midlife Drp1 to improve mitochondrial respiratory function is dependent upon Atg1 (Fig. 7F-H). **Critically, we now also show that the ability of midlife Drp1 to improve proteostasis is dependent upon Atg1 (Fig. 7I-L).** Thus, the ‘molecular link’ between midlife induction of mitochondrial fission and improved mitochondrial homeostasis/proteostasis is Atg1. Indeed, we now include data showing that promoting Drp1-mediated mitochondrial fission in Atg1 depleted animals leads to increased levels of ubiquitinated proteins in the mitochondrial fraction from aged muscle tissue (Fig. 7K-L).

Most of the mitochondrial analyses are focused in flight muscle, but the anti-aging effect of Drp1 manipulation occurs in neurons or intestine. It is therefore important to show that what is observed in flight muscle applies to neurons or intestinal cells.

This revised manuscript includes data from brain tissue showing that midlife induction of Drp1 promotes mitochondrial fragmentation (Fig. 1C), improves mitochondrial OXPHOS function, ETS capacity and complex I activity (Fig. S4B) and improves proteostasis (Fig. 5k-n).

To further test the idea that midlife Drp1 induction promotes lifespan by promoting mitophagy, the experiment should be performed in PINK1 or Parkin mutant condition, where mitophagy is blocked.

We understand the rationale for the suggested experiment. However, for a number of reasons, the suggested experiment would not result in data that could be interpreted in a meaningful manner:

- 1) Pink1 and Parkin mutants have dramatically shortened lifespan. More specifically, parkin mutants have a mean lifespan of ~7 days. We don’t think that it would be meaningful (or technically feasible) to upregulate Drp1 in ‘middle-aged’ (3 day old?) parkin mutants.**
- 2) Functional defects, including mitochondrial defects, in Pink1 mutants can be rescued by Parkin expression (*Nature* 441, 1157-1161; *Nature* 441, 1162-1166 (2006), indicating that mitophagy is not ‘blocked’ in Pink1 mutants. Mitochondrial defects in pink1/parkin mutants can also be rescued by Mitochondrial ubiquitin ligase 1 (MUL1) expression (*eLife*. 2014; 3: e01958), again suggesting that ‘mitochondrial homeostasis’ can be restored in pink1/parkin mutants.**
- 3) Moreover, it has been shown that pink1/parkin mutant flies display reduced mitochondrial fission/increased fusion. One interpretation of these findings is that “the loss of mitochondrial and tissue integrity in *PINK1* and *parkin* mutants derives from reduced mitochondrial fission.”. Critically, it has also been shown that Drp1 overexpression can rescue functional defects in pink1 and parkin mutants:**

<http://www.pnas.org/content/105/38/14503.long>
<http://www.pnas.org/content/105/5/1638.long>

To be clear, promoting mitochondrial fission in pink1/parkin mutants can restore mitochondrial integrity and rescue associated physiological defects. While many questions remain, it would seem that Drp1 can ‘bypass’ pink1/parkin to restore mitochondrial homeostasis in these mutants.

In summary, the suggested experiment is not technically feasible and would not directly test the idea that “*midlife Drp1 induction promotes lifespan by promoting mitophagy*”

Minor comments:

Is the pro-longevity effect of Marf RNAi also ATG dependent?

Yes, we now show that the pro-longevity effect of Marf RNAi is also dependent on Atg1 (Fig. S7c). Thanks for the suggestions, which have helped to improve our paper.

Reviewer #2 (Remarks to the Author):

Rana and col. investigate a highly relevant topic: the role of mitochondrial dynamics in the rate of ageing. The authors describe that old flies have more elongated (hyper-fused) mitochondria, which they claim correlates with a reduction in mitochondrial function. Next, they show that up-regulation of Drp1 (a protein essential for mitochondrial fission) extends lifespan when it is overexpressed in midlife. Interestingly, short-term induction of Drp1 (7 days) elicits the same effect. Finally, the authors claim that the lifespan extension reported requires the autophagy machinery of the cell.

I find the manuscript very interesting and well written, however some few experiments are required to back up the main conclusions of the work and some controls are missing. It seems to me that there are some mistakes in the way results are represented in the figures (see below).

Major comments

(1) There is one relevant lifespan experiment missing. What are the consequences of overexpressing Drp1 continuously during lifespan? Since there is no negative effect when Drp1 is expressed early in life, I predict that continuous expression should extend lifespan. If that is not the case, experiments to check the effect of continuous Drp1 expression on mitochondrial function in old flies (37 days) need to be produced.

We have now carried out multiple trials of the suggested experiment. Interestingly, we observe continuous overexpression of Drp1 in adult flies confers a very modest extension of lifespan (~5%) in two out of three trials and no effect in one of the three trials. As suggested, we assayed mitochondrial function in aged (37 day old) flies following

continuous expression of Drp1 (data provide below). Consistent with the modest/no effect on lifespan, there was no significant overall improvement in mitochondrial function:

(2) The data shown suggest that hyper-fusion of mitochondria in old age is deleterious and contribute to ageing and age-associated deleterious changes. However, this somehow contradicts papers showing that mitochondrial fusion is a mechanism of protection against mitochondrial damaged. To clarify the role of mitochondrial fusion in the context of ageing, the authors can overexpress pro-fusion proteins such as Marf or Opa1 in young or middle age flies and see which is the effect on lifespan. If fusion is deleterious lifespan will be shorten.

We thank the reviewer for this helpful suggestion. We now include data showing that overexpression of Marf (Mfn) in midlife does indeed shorten lifespan (Fig. S1i). Continuous overexpression of Mfn (i.e. starting in young flies) also shortens lifespan (see below). Finally, we additionally now include data to show that inhibiting mitochondrial fission in midlife, by expression of a Drp1-dominant negative transgene, also shortens lifespan (Fig S1h). Taken together, our data support the model in which a midlife shift towards increased fusion/decreased fission contributes to tissue and organismsal aging.

(3) Which is the effect of specific overexpression of Drp1 in muscle? Since most of the data about mitochondrial function have been performed in muscle, this experiment should be included. If lifespan is not extended assays of mitochondrial function in gut and/or brain should be performed.

For technical reasons (there is no muscle-specific inducible Gene-Switch driver; MHC-GS shows high levels of RU486-independent expression and RU486-dependent expression in the intestine), we are not able to test the effect of midlife induction exclusively in muscle. However, as requested, we now show improved mitochondrial OXPHOS function, ETS capacity, complex I activity in brain tissue following midlife Drp1 induction (Fig. S4b).

(4) Is the effect of Drp1 mediated through canonical mitophagy via Pink1/Parkin? If the authors have data of the effects on lifespan of overexpressing Drp1 on a background where Pink1 and Parkin are depleted, I would strongly recommend them to include these data in the manuscript. However, this experiment is not absolutely necessary to support the main conclusions of the paper.

We thank the reviewer for recognizing that this experiment is not “*necessary to support the main conclusions of the paper*”.

(5) Some controls are missing:

a. Figure 4a-d. Young controls are required to show that CI activity and mitochondrial respiration are decreased and ROS are increased in old individuals, and that these phenotypes are rescued by Drp1 expression. Why did the authors choose 44d for figure b? They don't use this age in any other experiment and don't show data of mitochondrial fission/fusion levels at this age.

We now show Complex I, respiration and ROS assays in young flies (Fig. S4a-c). For Figure 4b, we included the 44 day time point to demonstrate that midlife induction of Drp1 (from 30 days), leads to long-term improvement in mitochondrial function.

b. Figure 5f-j. Similarly, the authors should show young controls.

We now show young controls.

c. Figure S3C-E young controls are missing.

Here, the experiment is shown to demonstrate that RU486 doesn't impact mitochondrial morphology or function in control flies (daGS>w). Since RU486 is provided from day 30-37, there is no 'young time-point' with/without RU486.

d. Figures S4a-c young controls are missing.

Here, the experiment is shown to demonstrate that RU486 doesn't impact mitochondrial complex I activity or ROS levels in control flies (daGS>w). Since RU486 is provided from day 30-37, there is no 'young time-point' with/without RU486.

(6) Figure 7. The authors show that Drp1 overexpression rescues the elongation phenotype associated with age in an Atg1 depleted background, but lifespan is not extended. Is it mitochondrial respiration and complex I activity increased? Are ROS reduced? These experiments should be performed to demonstrate that mitochondrial turnover is not working when Atg1 is knock-down.

Thanks. This was a great suggestion that we were planning to carry out. We have now done so and the findings are presented in Fig. 7f-h. Importantly, Atg1 is required for Drp1-mediated improved mitochondrial respiration, ETS capacity, complex I activity and reduced ROS levels! In addition, we also show that Atg1 is required for Drp1-mediated improved proteostasis (Fig. 7i-l)! We agree that this data significantly improves the paper.

(7) How many times have been the lifespan experiments repeated? Since differences are small in some experiments, lifespan studies should have at least 2-3 repetitions. A summary of the lifespan data can be presented as in a former paper from the same group: Ulgherait et al. Cell Reports. 2014.

All lifespan experiments have been carried out at least two times. As suggested, we now include a Table detailing all lifespan experiments (Supp. Table 1).

(8) Data are presented as mean \pm SEM in most figures, however some of the figures do not represent the mean or SEM e.g. 1b or 2f. Please check this.

Thanks. We have checked this and when indicated data do represent mean +/- SEM

Minor comments:

(1) Are blots represented in figures 6a and S6a the same? Only one blot should be shown (better S6a). In addition, quantification in 6b should be done for each specific protein.

Yes, they are the same. We think it is helpful to the reader to show the entire blot in S6a. As suggested, we have now quantified each specific protein in 6b.

(2) Figure 7. Protein or mRNA levels of Atg1 should be shown to show that the knock-down is working.

We have now validated the Atg1 mRNA is indeed reduced upon RNAi (Fig. 7a).

(3) The authors should cite those other papers that show different results than the ones observed

by them, for example the work of the Osiewacz's lab in yeast, several papers in *C. elegans* (e.g. Regmi and col. 2014. *Aging and Yang and col. 2011. Aging Cell*). These papers do not contradict their work, but indicate different mechanisms working in different species.

Thanks. We have now done so. The major difference between our work and those studies is that we manipulated mitochondrial dynamics in (middle-) aged animals.

Reviewer #3 (Remarks to the Author):

The manuscript entitled "Promoting Drp1-mediated mitochondrial fission in midlife prolongs healthy lifespan" from Rana A. et al. describes that transient induction of the dynamin-related protein Drp1 in midlife of fruit flies decelerates their aging and increases their fitness. The data shown in the manuscript are very robust, and the experiments are well-done. The findings are unexpected, when comparing with the findings from other model systems where Drp1 plays a detrimental role (see below). The manuscript proposes a paradigm of aging with a protective role of mitochondrial fragmentation. Due to its potential relevance for mammalian systems, the topic of this manuscript is of very high interest for the general readership of Nature Communications. However, before being considered as manuscript to be published in this journal, the authors have to overcome some major concerns and deal with minor improvements as listed below.

Major concerns:

1.) The reader of the manuscript may get the impression that it is to be expected that mitochondrial fragmentation is beneficial for cells, i.e. increased health and increased lifespan. But this is not true. Deletion of the yeast and worm homologues of Drp1 resulted in increased lifespans, and not in decreased lifespans (e.g., Scheckhuber CQ et al., Nat Cell Biol 2007, PMID: 17173038; Yang CC et al., Aging Cell 2011, PMID: 21463460). Similarly, inhibiting Drp1 expression may be protective in some neurodegenerative scenarios (e.g., Manczak M et al., Hum Mol Genet 2016, PMID: 27677309; Kandimalla R et al., Hum Mol Genet 2016, PMID: 27634647). In contrast, in mammalian cells in vitro and in vivo knock-out of Drp1 resulted in severe loss of survival leading to neurodegeneration (e.g., Kageyama Y et al., J Cell Biol 2012). This could be the result of the accumulation of damaged mitochondria upon defective mitophagy, whose efficiency depends (in some model systems) on mitochondrial fragmentation. The findings of the authors favor the second concept (which is therefore not very novel). However, the authors have to mention and discuss the discrepancies, which can be found in the literature. Similarly, it has to be noted that mitochondrial fragmentation is not in all cases required for mitophagy, and that mitochondrial fragmentation does not in all cases trigger mitophagy. One might get the impression when reading the manuscript that these things are obvious. Having the discrepancies of protective/detrimental roles of Drp1 in mind, I strongly recommend further substantiating the protective role of Drp1 for health and lifespan in fruit flies. I recommend expressing dominant-negative Drp1 mutants in fruit flies at the very same conditions as wild-type Drp1 (at least for one or two key experiments). If the model of the authors is true, life span and health should decline when expressing dominant-negative Drp1 mutants, whereas expressing wild-type Drp1 is

protective. Alternatively, the authors may perform a knock-down of Drp1.

We agree that papers investigating the role of mitochondrial dynamics in aging/lifespan in yeast and worms should be considered, discussed and cited and we have now done so. We have now corrected our failure to cite this important work.

The major difference between our work and the previous studies is that we have manipulated mitochondrial dynamics in AGED ANIMALS. We think that this is key.

We are very grateful for the suggestion to express dominant negative Drp1 (Drp1-DN) in middle-aged flies. We have now carried out this experiment and we find that midlife induction of Drp1-DN SHORTENS lifespan (Fig. S1h). Moreover, we have additionally examined the impact of midlife induction of Mitofusin (Mfn). We now show that up-regulation of Mfn in midlife also shortens lifespan (Fig. Si). This suggestion was very helpful towards the goal of strengthening our findings and improving our manuscript. Thank you.

2.) Drp1 triggers mitochondrial fragmentation, but it is also critically involved in the division of peroxisomes. From my point of view, it cannot be excluded that increased peroxisomal fission is the protective aspect in the scenarios describes by the authors. Promoting peroxisomal fission might lead to reduced ROS levels, thereby reducing the damage of mitochondria, increasing health and lifespan. I've noticed that silencing the mitochondrial fusion factor dMfn1 had similar protective effects than expressing Drp1. Therefore, the mitochondrion-based scenario is likely, and no additional experiments are needed, but the authors should at least write that a role of peroxisomes on health- and lifespan prolongation cannot be excluded.

Thanks. We have now done so.

3.) The finding that Drp1 induction in midlife reduces the levels of ubiquitylated aggregates, which co-migrate with mitochondrial fractions, is very fascinating (for me the most fascinating aspect of the paper at all). This raises the question of the protein species, which are ubiquitylated. I could imagine that these are mitochondrial precursor proteins, which get ubiquitylated, if they cannot be imported properly into mitochondria. Mitochondrial protein import depends on mitochondrial membrane potential, which is obviously markedly reduced in fruit flies upon aging (see Fig. 3k). Thus, Drp1-mediated mitochondrial fission may restore the mitochondrial import machinery, preventing the detrimental accumulation of mitochondrial precursor protein in the cytosol. The aberrant accumulation of mitochondrial precursor proteins has been reported very recently in different model systems (Wang X et al., Nature 2016, PMID: 26192197; Wrobel L et al., Nature 2016, PMID: 26245374). I strongly recommend enriching ubiquitylated proteins from mitochondrial fractions, and sending them to MS for identification. If you see the enrichment of mitochondrial precursor proteins in the cytosol upon aging, which is reduced when Drp1 is expressed, this would favor the model as described above.

Thanks again. This is a helpful suggestion for future work. However, we hope that you agree that carrying out this type of experimental work which would be new to us, in a careful fashion, is beyond the scope of the current manuscript.

To provide further insight into the molecular link between mitochondrial morphology and proteostasis, we have carried out a significant amount of new experimental work. In this revised manuscript, we now include data to show that expression of a dominant-negative Drp1 (Drp1-DN) transgene in midlife impairs proteostasis (Fig. 5e-h). Thus, strengthening our previous data showing that midlife Drp1-mediated mitochondrial fission improves proteostasis in aged flies. Furthermore, we now show that up-regulating Mfn in midlife impairs proteostasis (Fig. S5c,d). Taken together, our data supports a model in which a midlife shift towards a more elongated mitochondrial morphology in flight muscle impairs the clearance of damaged mitochondria/mitochondrial proteins via mitophagy. We show that promoting mitochondrial fission in midlife facilitates mitophagy and/or promotes the turnover of some mitochondrial proteins (Fig. 6). We hypothesize that facilitating mitophagy (by promoting fission) in midlife reduces the levels of damaged/insoluble (mitochondrial) proteins in aged muscle tissue. Indeed, it has been reported that there is an over-representation of mitochondrial proteins in the insoluble protein fraction from aged animals^{1,2}.

3.) How do you explain that Drp1 expression reduces the levels of some respiratory chain complexes (see Fig. 6a) and the finding that Drp1 expression increases ETS and OXPHOS activities (see Fig. 4)? I am puzzled about these counterintuitive results... I've realized that you have discussed that some respiratory chain complexes might be selectively targeted to mitophagy, but why should this enable increase respiratory chain activities?

Our working hypothesis is that promoting Drp1-mediated fission in midlife facilitates the selective clearance of damaged mitochondria/mitochondrial proteins/mitochondrial components. There are a number of possible explanations why the selective turnover of damaged mitochondrial components/proteins could result in improved respiratory function at the tissue level. For example, it is possible that the accumulation of dysfunctional mitochondria has a detrimental ‘contagion effect’ on healthy mitochondria in the tissue. Another plausible explanation is the idea that within individual mitochondria the accumulation of damaged proteins/components interferes with the assembly and/or activity of the OXPHOS machinery; hence, selective clearance of damaged mitochondrial components could improve OXPHOS activity.

Interestingly, there is emerging evidence to support a model in which mitochondrial fission can lead to the ‘segregation of damaged mitochondrial components’, ie uneven daughter units in terms of mitochondrial membrane potential: *Circ Res.* 2015 May 22; 116(11): 1835–1849.

4.) The authors have shown markers of mitophagy upon expression of Drp1 (see Fig. 6), and suggest that increased mitophagy upon Drp1-dependent mitochondrial fragmentation is

beneficial, because it reduces the number of damaged mitochondria. I am not yet convinced whether this is true. The increased recruitment of Atg8a/LC3 to mitochondria (see Fig. 6g+h) could be a result of general autophagy induction upon Drp1 expression, which leads to more Atg8a/LC3 dots in general that therefore co-localize more often with mitochondria. I am not familiar with mitophagy in fruit flies, but if there is a mitophagy-specific factor know that upon knock-down interferes with autophagy, the authors should knock-down this factor (interfering with mitophagy), in order to prove that mitophagy is the beneficial pathway required for the protective effect of Drp1 expression. If this is not possible, the authors should at least state that their data suggest that mitophagy is the protective effect, they shouldn't write that their data demonstrate or indicate to a crucial role of mitophagy. This statement is too strong.

Unfortunately, there is no 'mitophagy-specific' factor yet characterized in flies. We thank the reviewer for their advice regarding the interpretation of the findings. We believe that, in this revised manuscript, we are very careful not to over interpret our data.

5.) The authors demonstrate that general autophagy is needed for the protective effect of Drp1 (see Fig. 7). I am wondering why this is the case. The authors suggest that mitophagy is pivotal here, which might be the case (see 4.). But I think that induction of general autophagy could also reduce the levels of (mistargeted) ubiquitylated (mitochondrial) proteins in the cytosol (see Fig. 5). I strongly recommend analyzing the levels of ubiquitylated proteins in cells expressing Drp1 but lacking Atg1. Drp1 (similar to Fig. 5). If Drp1 triggers autophagy leading to a decrease of damaged mitochondria, and in parallel to a decrease in ubiquitylated proteins in the cytosol, this would be very fascinating.

Thanks. This was a great suggestion that we were planning to carry out. We have now done so and the findings are presented in Fig. 7i-l. In addition, we have now also included new data to show that Atg1 is required for Drp1-mediated improved mitochondrial respiration, ETS capacity, complex I activity and reduced ROS levels (Fig. 7f-h). Interestingly, we now show that promoting Drp1-mediated mitochondrial fission in Atg1 depleted animals impairs proteostasis and leads to increased levels of ubiquitinated proteins in the mitochondrial fraction from aged muscle tissue.

Our working hypothesis is that the accumulation of protein aggregates in middle-aged flight muscle results from a failure in mitophagy. Our data support the model that the failure in mitophagy results from a shift in mitochondrial dynamics (reduced fission).

6.) The information about the statistical tests used in this manuscript is insufficient. A paragraph dealing with this information must be included in the Material & Methods section. The Student's t-test (unpaired or paired? One-tailed or two-tailed?) used in most cases is only adequate if comparing two groups of data. Have you used suitable pre-tests, such as equal variance and normality tests, in order to test whether you are allowed to use the Student's t-test at all? If you have three or more groups as depicted your graphs other statistical tests, such as ANOVA are needed, and the pairwise comparisons should then be performed afterwards using suitable posthoc tests (Holm-Sidak, Tukey or equivalents). Similar to Student's tests, suitable pre-tests must be performed to test whether ANOVA can be used at all. Please define for every

experiment what you mean with “n”: The number of independent experiments? The number of biological replicates within one or several independent experiments? The number of flies?

Many thanks. We have now done so.

Minor points:

1.) Introduction (page 3, lanes 61-62): “mitochondrial outer membrane proteins” instead of “outer mitochondrial membrane proteins” (for consistency)

2.) Discussion (page 11, lines 239-240): Drp1 induction correlates with decreased dMfn levels, it is not clear whether there is a causative connection between to observations.

3.) Discussion (page 12, lines 262-264): These findings SUGGEST, they do not indicate (that’s too strong)

4.) Discussion (page 12): Please refer to Fig. 7b

5.) Supplementary data (pages 10/11): “radicals” instead of “radicles”

Many thanks. We have now made the suggested changes.

References:

- 1 David, D. C. *et al.* Widespread protein aggregation as an inherent part of aging in *C. elegans*. *PLoS Biol* **8**, e1000450, doi:10.1371/journal.pbio.1000450 (2010).
- 2 Reis-Rodrigues, P. *et al.* Proteomic analysis of age-dependent changes in protein solubility identifies genes that modulate lifespan. *Aging Cell* **11**, 120-127, doi:10.1111/j.1474-9726.2011.00765.x (2012).

Reviewers' Comments:

Reviewer #1:

Remarks to the Author:

The authors have adequately addressed the comments from this reviewer. The manuscript is now recommended for publication.

Reviewer #2:

Remarks to the Author:

The authors have addressed all my comments and concerns, and have greatly improved the manuscript, which now clearly supports their claims.

I find very intriguing the fact that continuous overexpression of Drp1 doesn't extend fly lifespan as expected. Accordingly mitochondrial function is not improved. These results indicate that overexpression of Drp1 triggers some "unknown" changes that prevent the positive effects associated with midlife overexpression. Obviously, it is out of the scope of the present manuscript to figure out why this occurs, however this data must be included in the manuscript and discussed in the appropriate section(s) of the paper.

The inclusion of these results (including the whole life expression of dMfn) did not affect the quality of the work or the validity of the conclusions, and will greatly help other researchers to replicate the work done by the authors, and plan new experiments to understand this intriguing observation.

Finally, I would like to congratulate the authors for a fantastic work.

Reviewer #3:

Remarks to the Author:

The authors adequately addressed all the major concerns and minor comments. They included an impressive amount of new data, which support their model that the mitochondrial fission factor Drp1 prolongs healthy lifespan in flies when expressed in midlife. Most of all, the data using a dominant-negative form of Drp1, which impaired with healthy lifespan elongation are very convincing. The same is true for the data showing that the beneficial effects of Drp1 on mitochondrial clearance and turnover of ubiquitylated mitochondrial proteins depend on ATG1. Therefore, I strongly encourage the publication of this very valuable contribution in Nature Communications.

Reviewer #1 (Remarks to the Author):

The authors have adequately addressed the comments from this reviewer. The manuscript is now recommended for publication.

Thanks

Reviewer #2 (Remarks to the Author):

The authors have addressed all my comments and concerns, and have greatly improved the manuscript, which now clearly supports their claims.

I find very intriguing the fact that continuous overexpression of Drp1 doesn't extend fly lifespan as expected. Accordingly mitochondrial function is not improved. These results indicate that overexpression of Drp1 triggers some "unknown" changes that prevent the positive effects associated with midlife overexpression. Obviously, it is out of the scope of the present manuscript to figure out why this occurs, however this data must be included in the manuscript and discussed in the appropriate section(s) of the paper.

The inclusion of these results (including the whole life expression of dMfn) did not affect the quality of the work or the validity of the conclusions, and will greatly help other researchers to replicate the work done by the authors, and plan new experiments to understand this intriguing observation.

Finally, I would like to congratulate the authors for a fantastic work.

Thanks. We now include this data in the manuscript.

Reviewer #3 (Remarks to the Author):

The authors adequately addressed all the major concerns and minor comments. They included an impressive amount of new data, which support their model that the mitochondrial fission factor Drp1 prolongs healthy lifespan in flies when expressed in midlife. Most of all, the data using a dominant-negative form of Drp1, which impaired with healthy lifespan elongation are very convincing. The same is true for the data showing that the beneficial effects of Drp1 on mitochondrial clearance and turnover of ubiquitinated mitochondrial proteins depend on ATG1. Therefore, I strongly encourage the publication of this very valuable contribution in Nature Communications.

Thanks